# Towards a satellite formaldehyde – in situ hybrid estimate for organic aerosol abundance

Jin Liao[1,2], Thomas F. Hanisco[1], Glenn M. Wolfe[1,3], Jason St. Clair[1,3], Jose L. Jimenez[4,5], Pedro Campuzano-Jost[4,5], Benjamin A. Nault[4,5], Alan Fried[6], Eloise A. Marais[7,*], Gonzalo Gonzalez Abad[8], Kelly Chance[8], Hiren T. Jethva[1,2], Thomas B. Ryerson[9], Carsten Warneke[9,5], Armin Wisthaler[10,11]

[1]Atmospheric Chemistry and Dynamic Laboratory, NASA Goddard Space Flight Center, Greenbelt, MD, USA
[2]Universities Space Research Association, GESTAR, Columbia, MD, USA
[3]University of Maryland Baltimore County, Joint Center for Earth Systems Technology, Baltimore, MD, USA
[4]Department of Chemistry, University of Colorado, Boulder, Colorado, USA
[5]Cooperative Institute for Research in the Environmental Sciences, University of Colorado, Boulder, Colorado, USA
[6]Department of Atmospheric and Oceanic Sciences, University of Colorado Boulder, Boulder, Colorado, USA
[7]School of Geography, Earth and Environmental Sciences, University of Birmingham, UK
[8]Harvard-Smithsonian Center for Astrophysics, Cambridge, Massachusetts, USA
[9]NOAA Earth System Research Laboratory (ESRL), Chemical Sciences Division, Boulder, CO, USA
[10]Department of Chemistry, University of Oslo, Oslo, Norway
[11]Institute for Ion Physics and Applied Physics, University of Innsbruck, Innsbruck, Austria
[*]now at Department of Physics and Astronomy, University of Leicester, Leicester, UK

Correspondence email: jin.liao@nasa.gov

**Abstract**

Organic aerosol (OA) is one of the main components of the global particulate burden and intimately links natural and anthropogenic emissions with air quality and climate. It is challenging to accurately represent OA in global models. Direct quantification of global OA abundance is not possible with current remote sensing technology; however, it may be possible to exploit correlations of OA with remotely observable quantities to infer OA spatiotemporal distributions. In particular, formaldehyde (HCHO) and OA share common sources via both primary emissions and secondary production from oxidation of volatile organic compounds (VOCs). Here, we examine OA-HCHO correlations using data from summer time airborne campaigns investigating biogenic (NASA SEAC$^4$RS and DC3), biomass burning (NASA SEAC$^4$RS) and anthropogenic conditions (NOAA CalNex and NASA KORUS-AQ). In situ OA correlates well with HCHO (r = 0.59 – 0.97), and the slope and intercept of this relationship depend on the chemical regime. For biogenic and anthropogenic regions, the OA-HCHO slopes are higher in low $NO_x$ conditions, because HCHO yields are lower and aerosol yields are likely higher. The OA-HCHO slope of wild fires is over 9 times higher than that for biogenic and anthropogenic sources. The OA-HCHO slope is higher for highly polluted anthropogenic sources (e.g., KORUS-AQ) than less polluted (e.g., CalNex) anthropogenic sources. Near-surface OA over the continental US are estimated by combining the observed in situ relationships with HCHO column retrievals from NASA's Ozone Monitoring Instrument (OMI). HCHO vertical profiles used in OA estimates are from climatology a-priori profiles in the OMI HCHO retrieval or output of specific period from a newer version of GEOS-Chem. Our OA estimates compare well with US EPA IMPROVE data obtained over summer months

(e.g., slope = 0.60-0.62, r = 0.56 for August 2013), with correlation performance
comparable to intensively validated GEOS-Chem (e.g., slope = 0.57, r = 0.56) with
IMPROVE OA and superior to the satellite-derived total aerosol extinction (r = 0.41)
with IMPROVE OA.  This indicates that OA estimates are not very sensitive to these
HCHO vertical profiles and that a priori profiles from OMI HCHO retrieval have a
similar performance to that from the newer model version in estimating OA. Improving
the detection limit of satellite HCHO and expanding in situ airborne HCHO and OA
coverage in future missions will improve the quality and spatiotemporal coverage of our
OA estimates, potentially enabling constraints on global OA distribution.

## 1. Introduction

Aerosols are the largest source of uncertainty in climate radiative forcing (IPCC 2013; Carslaw et al., 2013) and decrease atmospheric visibility and impact human health (Pope 2002). Organic aerosols (OA) comprise a large portion (~50%) of submicron aerosols (Jimenez et al., 2009; Murphy et al., 2006; Shrivastava et al., 2017), and this fraction will grow with continued decline in $SO_2$ emissions (Attwood et al., 2014; Marais et al., 2017; Ridley et al., 2018). In addition, OA serve as cloud condensation nuclei (CCN) and affect cloud formation and climate radiative forcing. OA components also have adverse health effects (e.g., Walgraeve et al., 2010) and contribute significantly to regional severe haze events (e.g., Hayes et al., 2013). Finally, because the response of temperature to changes in climate forcing is non-linear (Taylor and Penner, 1994) and the forcing by aerosols has strong regional character (Kiehl and Briegleb, 1993), it is necessary to separate out different climate forcing components to accurately forecast the climate response to changes in forcing.

Despite their importance, it has been challenging to accurately represent OA in global models. Chemical transport models (CTMs) often under-predict OA (e.g., more than a factor of 2 lower OA near the ground) compared to observations, and model-to-model variability can exceed a factor of 100 in the free troposphere (Tsigaridis et al., 2014; Heald et al., 2008; Heald et al., 2011). Fully explicit mechanisms have attempted to capture the full OA chemical formation mechanisms (e.g., Lee-Taylor et al., 2015), but it is too computationally expensive to apply these mechanisms to OA formation in global CTMs at a useful resolution. For computational efficiency, 3-D models such as GEOS-

Chem include direct emissions of primary OA (POA) and represent secondary OA (SOA)
formation either by lumping SOA products according to similar hydrocarbon classes
(Kim et al., 2015) or based on the volatility of the oxidation products (Pye et al., 2010).
Marais et al. (2016) applied an aqueous phase mechanism for SOA formation from
isoprene in GEOS-Chem to reasonably simulate isoprene SOA in the southeastern (SE)
US. Schroder et al. (2018) showed GEOS-Chem has a very large under prediction of
SOA in the Northeastern US dominated by anthropogenic emissions. Accurate emission
inventories are also needed to correctly represent volatile organic compounds (VOCs)
and $NO_x$ ($NO_x$ = NO + $NO_2$) inputs, and these often have biases compared to
observational constraints (Kaiser et al., 2018, Travis et al., 2016, Anderson et al., 2014;
McDonald et al., 2018).

A quantitative measure of OA from space would be helpful for verifying emissions and
aerosol processes in models. However, direct measurements of OA from space are
currently unavailable. Aerosol optical depth (AOD) measured by satellite sensors
provides a coarse but global picture of total aerosol distributions. Multi-angle Imaging
SpectroRadiometer (MISR) provides aerosol property information such as size, shape and
absorbing properties, which allows retrieving the AOD of a subset of aerosols (Kahn and
Gaitley, 2015). Classification algorithms have been developed to speciate different
aerosol types (e.g., OA) based on AOD, extinction Angstrom exponent, UV Aerosol
Index, and trace gas columns from satellite instruments (de Vries et al., 2015). Here we
aim to provide a quantitative estimation of OA mass concentrations from satellite
measurements.

Formaldehyde (HCHO) is one of the few VOCs that can be directly observed from space. Sources emitting POA (e.g., biomass burning (BB)) often simultaneously release VOCs. HCHO and SOA are also both produced from emitted VOCs. VOCs, as well as intermediate- and semi- volatile organic compounds (I/SVOCs), are oxidized by hydroxyl radicals (OH) to form peroxy radicals ($RO_2$), which then react with NO, $RO_2$, or hydroperoxy radicals ($HO_2$) or isomerize. These oxidation processes produce HCHO and oxidized organic compounds with low volatility that condense to form SOA (Robinson et al., 2013; Ziemann and Atkinson, 2012). The yield of HCHO and SOA from hydrocarbon oxidation thus depends on the VOC precursors, oxidants (OH, $O_3$ and $NO_3$), $RO_2$ reaction pathway (e.g., NO levels), and pre-existing aerosol abundance and properties (Wolfe et al., 2016; Pye et al., 2010; Marais et al., 2016 and 2017; Xu et al., 2016). Moreover, although the lifetime of HCHO (1-3 hrs) is shorter than OA (1 week), HCHO continues to form from slower reacting VOCs, as well as from the oxidation of later generation products. Observations across megacities around the world show that OA formation in polluted/urban area happens over about 1 day (e.g., DeCarlo et al., 2010; Hodzic and Jimenez, 2011; Hayes et al., 2013; 2015), and HCHO is also significantly formed over this timescale (Nault et al., 2018). In addition, Veefkind et al. (2011) found that satellite AOD correlated with HCHO over the summer time SE US, BB regions, and Southeast Asian industrialized regions. This also suggests that OA share common emission sources and photochemical processes with HCHO and are a major contributor to AOD in the regions above. Marais et al. (2016) further used the relationship between aircraft OA and

satellite HCHO to evaluate GEOS-Chem representation of SOA mass yields from
biogenic isoprene in the SE US.

We present an OA surface mass concentration estimate (OA estimate) derived from a
combination of satellite HCHO column observations and in situ OA-HCHO relationships.
Because the detection limit of satellite HCHO column observations limit the quality of
OA estimate, we focus our analyses on summer time when HCHO levels are high. The
OA estimate is evaluated against OA measurements at ground sites. A 3-D model GEOS-
Chem OA simulation is shown for comparison.

**2. Methods**
**2.1 In situ airborne observations**
Figure 1 shows flight tracks with altitudes < 1 km of the field campaigns used in the
current study. The Studies of Emissions, Atmospheric Composition, Clouds and Climate
Coupling by Regional Surveys (SEAC[4]RS) mission (Toon et al., 2016) covered the
continental US with a focus on the SE US in August-September 2013. The Deep
Convective Clouds & Chemistry Experiment (DC3) (Barth et al., 2015) surveyed the
central and SE US in May-June 2012, targeting isolated deep convective thunderstorms
and mesoscale convective systems. The California Research at the Nexus of Air Quality
and Climate Change (CalNex) (Ryerson et al., 2013) investigated the California region in
May-June 2010, targeting the Los Angeles (LA) Basin and Central Valley. The Korea-
United States Air Quality Study (KORUS-AQ) studied South Korean air quality,
sampling many large urban areas in South Korea and continental Asian outflow over the
West Sea, in May-June 2016 (Aknan and Chen, 2017). KORUS-AQ only includes data
with longitude < 133° E to exclude the transit from US because it targeted South Korea
and the nearby region. These field campaigns were selected as they had recent high-
quality in situ HCHO and OA data measured with state-of-the-art instruments and studied
summer time regional tropospheric chemical composition.

In situ airborne HCHO observations were acquired by multiple instruments. The DC3
NASA DC-8 payloads featured two HCHO measurements: the NASA In Situ Airborne
Formaldehyde (ISAF) (Cazorla et al., 2015) and the Difference Frequency Generation
Absorption Spectrometer (DFGAS) (Weibring et al., 2006). The SEAC$^4$RS NASA DC-8
payloads also featured two HCHO measurements: the NASA ISAF and the Compact
Atmospheric Multispecies Spectrometer (CAMS) (Richter et al., 2015). HCHO
measurements from ISAF were found to be in good agreement with CAMS, with a
correlation coefficient of 0.99 and a slope of 1.10 (Zhu et al., 2016). HCHO
measurements from ISAF also had a good agreement with DFGAS, with a correlation
coefficient of 0.98 and a slope of 1.07. Because ISAF has higher data density, we used
ISAF HCHO data for DC3 and SEAC$^4$RS. During KORUS-AQ, CAMS was the only
HCHO instrument onboard the DC-8. In CalNex a proton transfer reaction mass
spectrometer (PTR-MS) (Warneke et al., 2011) was used to measure HCHO on board the
NOAA P3 aircraft.

In situ airborne OA from SEAC$^4$RS, DC3, and KORUS-AQ was measured by the
University of Colorado High-Resolution Time-of-Flight Aerosol Mass Spectrometer
(AMS, DeCarlo et al., 2006; Dunlea et al., 2009; Canagaratna et al., 2007; Jimenez et al.,
2016) and in situ airborne OA from CalNex was measured by the NOAA Compact Time-
of-Flight Aerosol Mass Spectrometer (Drewnick et al., 2005; Canagaratna et al., 2007;
Bahreini et al., 2012). The OA measurements are from 1 min merge data and converted
from $\mu g$ $sm^{-3}$ (at 273 K and 1013 mbar) to $\mu g$ $m^{-3}$ under local T & P for each data point,
to be consistent with HCHO concentrations in $\mu g$ $m^{-3}$ or molec $cm^{-3}$ at local T & P.

Although NO modulates the $RO_2$ lifetime, and thus, the production of HCHO and SOA,
NO cannot be directly observed via remote sensing. Instead, $NO_2$ can be directly
observed in space by satellites, and because $NO_2$ represents typically ~80% (e.g.,
SEAC[4]RS and KORUS-AQ) of the boundary layer $NO_x$ concentrations during the
daytime, $NO_2$ can be used as a surrogate for daytime NO concentrations and oxidative
conditions around the globe. In situ airborne $NO_2$ was measured by the NOAA
Chemiluminescence $NO_yO_3$ instrument (Ryerson et al., 2001) during SEAC[4]RS, DC3,
and CalNex and by University of Berkeley laser induced fluorescence $NO_2$ instrument
(Day et al., 2002) during KORUS-AQ. SEAC[4]RS isoprene measurements were from
proton-transfer-reaction mass spectrometer (PTR-MS) (Wisthaler et al., 2002).

**2.2 Ground-based OA measurements**
Ground-based OA measurements over the US were from the EPA Interagency
Monitoring of Protected Visual Environments (IMPROVE) (Malm et al., 1994; Solomon
et al., 2014; Hand et al., 2014; Hand et al., 2013; Malm et al., 2017) and Southeastern
Aerosol Research and Characterization (SEARCH) (Edgerton et al., 2006) networks. In
the IMPROVE network, aerosols were collected on quartz fiber filters and analyzed in
the lab by thermal optical reflectance for organic and elemental carbon. The data were
reported every three days from 1988 to 2014. Monthly averages were used for
comparison in this study. IMPROVE OA data over the SE US (east of 70ºW) in summer
time were multiplied by a factor of 1.37 to correct for partial evaporation during filter
transport, following the recommendation of a comparison study with SEARCH organic
carbon (OC) measurements (Kim et al., 2015; Hand et al., 2013). Although IMPROVE
OA corrected for evaporation has potential uncertainties with the constant scaling factor,
the IMPROVE measurements have high spatial coverage. SEARCH network (Edgerton
et al., 2006; Hidy et al., 2014) OC was determined by the difference between total carbon
(TC) detected by a tapered element oscillating microbalance (TEOM) and black carbon
(BC) measured by an in situ Thermal Optical instrument. This allowed real-time
measurement of OC and prevented evaporation during filter transport. Although the
SEARCH network only has 5 sites available, we used observations from this network due
to their high accuracy. The IMPROVE and SEARCH network OC measurements were
converted to OA by multiplying by a factor of 2.1 based on ground and aircraft
observations (Pye et al., 2017; Schroder et al., 2018).

**2.3  Satellite measurements**
Satellite HCHO column observations were derived from the NASA's Ozone Monitoring
Instrument (OMI), a UV/Vis nadir solar backscatter spectrometer on the Aura satellite
(Levelt et al., 2006). Aura overpasses the equator at 1:30 pm local time, daily. Here we
used the OMI HCHO version 2.0 (collection 3) gridded ($0.25° \times 0.25°$) retrieval data
(Gonzalez Abad et al., 2015) from the Smithsonian Astrophysical Observatory (SAO).
Satellite data for HCHO columns were subjected to data quality filters: 1) solar zenith
angle lower than 70°, 2) cloud fraction less than 40%, and 3) main quality flag and the
xtrackquality flag both equal to zero (Harvard-Smithsonia Center for Astrophysics OMI
HCHO data product description). The monthly average HCHO columns were also
weighted by the column uncertainties of the pixels. The HCHO retrieval used a priori
profiles without aerosol information from the GEOS-Chem model (Gonzalez Abad et al.,
2015). Satellite $NO_2$ column observations were also derived from NASA's OMI level 3
data (Lamsal et al., 2014; NASA OMI $NO_2$ data archive). Satellite $NO_2$ observations
were used to calculate $NO_x$ related chemical factor dependent OA estimate (see Table 2).
Satellite AOD observations were acquired from the Moderate Resolution Imaging
Spectroradiometer (MODIS) onboard the Aqua satellite, using overpasses at about 1:30
pm local time. Here, we used collection 06 (NASA MODIS AOD data archive), retrieved
using the Dark Target (DT) and Deep Blue (DB) algorithms (Levy et al., 2015), monthly
average data.

**2.4 GEOS-Chem**
We used GEOS-Chem (v9-02) at 2° × 2.5° with 47 vertical layers to simulate HCHO and
OA globally, the same as that in Marais et al. (2016). GEOS-Chem was driven with
meteorological fields from the NASA Global Modeling and Assimilation Office
(GMAO). The OA simulation included POA from fires and anthropogenic activity and
SOA from the volatility-based reversible partitioning scheme (VBS) of Pye et al. (2010)
for anthropogenic, fire, and monoterpene sources, and an irreversible aqueous-phase
reactive uptake mechanism for isoprene. The aqueous-phase mechanism was coupled to
gas-phase isoprene chemistry and has been extensively validated using surface and
aircraft observations of isoprene SOA components in the SE US (Marais et al., 2016).
This model version used the fourth-generation global fire emissions database (GFED4)
(Giglio et al., 2013) as BB emission inventory. The model was driven with Goddard
Earth Observing System – Forward Processing (GEOS-FP) meteorology for 2013 and
sampled along the SEAC$^4$RS (2013) and KORUS-AQ (2016) flight tracks. The model
was also run with 10% decrease in biomass burning, biogenic, or anthropogenic
emissions as a sensitivity test to evaluate the contributions of different sources to OA and
HCHO budget. Model monthly mean surface layer OA and total column formaldehyde
were obtained around the OMI overpass time (12:00-15:00 local time) for 2008-2013
using Modern-Era Retrospective analysis for Research and Applications (MERRA)
(Gelaro et al., 2017) meteorology, as GEOS-FP was only available from 2012. This was
compared to the OA estimate derived from satellite HCHO.

Global isoprene emissions from the Model of Emissions of Gases and Aerosols from
Nature version 2.1 (MEGAN) (Guenther et al., 2006) and satellite $NO_2$ column data were
used to calculate an isoprene and $NO_2$ dependent OA estimate (see Table 2). Global
isoprene emissions from MEGAN were implemented in GEOS-Chem and driven with
MERRA (MEGAN-MERRA).

**2.5 Estimation of surface organic aerosol mass concentrations**
An estimate for surface OA mass concentration was calculated based on a simple linear
transformation.
$\varepsilon(\text{i}) = \Omega_{HCHO}(\text{i})\eta(\text{i})\alpha(i) + \beta(\text{i})$                                          Eq. (1)
Here, $\varepsilon(\text{i})$ is the OA estimate for grid cell i ($\mu g\ m^{-3}$), $\Omega_{HCHO}(\text{i})$ is the OMI HCHO column
density (molec $cm^{-2}$) in each $0.25° \times 0.25°$ grid cell (similar resolution to OMI HCHO
nadir pixel data), $\eta(\text{i})$ is the ratio of midday surface layer (~60 m) HCHO concentrations
(molec $cm^{-3}$) to column concentrations (molec $cm^{-2}$) from GEOS-Chem, and $\alpha(i)$ and
$\beta(\text{i})$ are the slope and intercept of a linear regression between OA and HCHO from low
altitude (<1 km) airborne in situ measurements. The in situ to column conversion factor
$\eta(\text{i})$ was similar to that used by Zhu et al. (2017) to convert HCHO columns into surface
concentrations. $\eta(\text{i})$ was derived from the HCHO a priori profiles used in SAO OMI air
mass factor (AMF) calculations (GEOS-Chem v9-01-03 climatology) or from GEOS-
Chem v9-02, which included updated isoprene scheme for OA and is the next version of
the model (v9-01-03) for a priori profiles used in SAO satellite HCHO retrievals. HCHO
a priori profiles were used to be consistent with satellite HCHO retrievals and also to
show that OA estimate can be derived without running a global model separately. The
newer version of GEOS-Chem was used to test the sensitivity of OA estimates to updated
version of $\eta$. The newer version of GEOS-Chem also allows sampling through the flight
tracks of a recent field campaign (SEAC[4]RS) and examining the factors impacting $\eta$ with
both modeled and measured HCHO profiles. The detailed information about the impact
of HCHO profiles on $\eta$ is provided in Sect. 5.

**2.6 Aerosol extinction from satellite measurements**
Currently remote sensing techniques observe aerosols by quantifying AOD. The MISR
satellite instrument can estimate a subset of AOD, using constraints on size range, shape
and absorbing properties, but it cannot distinguish OA from other submicron aerosol
compounds such as sulfate and nitrate and also requires AOD to be above 0.1. Because
MISR estimates a subset of AOD, it is discussed above to verify that we are not
neglecting a satellite dataset that has already captured OA AOD. Moreover, OA account
for a large and relatively constant fraction of submicron aerosols in the SE US (Kim et
al., 2015; Wagner et al., 2015) and are one of the major submicron aerosol components
over the US (Jimenez et al., 2009). Therefore, AOD was converted to extinction to
represent OA for comparison.
$A_{ext} = AOD(i)\delta(i)$                                                                    Eq. (2)
where $A_{ext}$ is the calculated aerosol extinction ($Mm^{-1}$), AOD(i) is aerosol optical depth
from MODIS (see Sect. 2.3) in each 0.25°×0.25° grid cell, and $\delta(i)$ ($m^{-1}$) is the ratio of
surface layer OA concentrations ($\mu g\ m^{-3}$, at ambient T & P) to column OA concentrations
($\mu g\ m^{-2}$) from GEOS-Chem multiplied by $10^6\ Mm^{-1}/m^{-1}$ . The shape of the average
vertical profile of OA (OA fraction: 0.54-0.7) was close to that of total aerosol mass over
SE US (Wagner et al., 2015) where a large fraction of the enhanced non-BB aerosol
concentrations in summer time over the US are located. Data with BB plumes
interferences were excluded in the following analysis. The potential contribution of dust
and nitrate could alter the shape of the vertical profiles and introduce uncertainties when
using OA vertical profiles for other parts of the US. However, the outliners in the aerosol
extinction compared to ground OA measurements (see Sec. 6.3) were not located outside
of the SE US. Similar vertical profile shapes of OA and submicron particles were also
observed in a campaign outside the US over South Korea (Nault et al., 2018). Although
OA accounted for ~40% of the total submicron particles, the shape of OA and total
submicron particles vertical profiles were nearly identical.

**3. In situ OA-HCHO relationship**
Although OA and HCHO share common VOC emission sources and photochemical
processes, their production rates from different emission sources and photochemical
conditions vary, as do their loss rates. We found the main factors that modulate OA-
HCHO relationships from in situ measurements and discussed in the following section.

**3.1 Regional and Source-Driven Variability**
For all regions and/or sources investigated, near-surface in situ OA and HCHO are well
correlated. A scatter plot of in situ OA vs. HCHO at low altitudes (<1 km) from a number
of field campaigns (SEAC$^4$RS, DC3, CalNex, and KORUS-AQ) is displayed in Fig. 2.
The slopes, intercepts, and correlation coefficients are summarized in Table 1. SEAC$^4$RS,
DC3, and CalNex excluded BB data when acetonitrile > 200 pptv (Hudson et al., 2004).
KORUS-AQ used a BB filter with higher acetonitrile (>500 pptv) because the air masses
with moderate acetonitrile enhancement (200-500 pptv) were actually from
anthropogenic emissions. This attribution is based on high levels of acetonitrile detected
downwind of Seoul and west coastal petrochemical facilities, the slope between
acetonitrile and CO being to urban emissions (Warneke et al., 2006), and the
concentrations of anthropogenic tracer CHCl$_3$ being high (Warneke et al., 2006). Similar
to acetonitrile, another common BB tracer hydrogen cyanide (HCN) was also enhanced
in these air masses. BB data (acetonitrile > 200 pptv) for SEAC$^4$RS were analyzed
separately and are inset in Fig. 2. Although all CalNex data had a tight correlation, we
only included the flight data near LA basin to target the area strongly influenced by
anthropogenic emissions. In general, the correlation coefficients between in situ OA and
HCHO were strong (r = 0.59 −0.97) (Table 1).
The variety in OA-HCHO regression coefficients among different campaigns reflects the
regional and/or source-driven OA-HCHO variability. Considering only the non-biomass
burning (non-BB) air masses sampled, OA and HCHO had the tightest correlation for
CalNex, because CalNex focused on the LA area (shown in Fig. 2) and Central Valley
while SEAC$^4$RS and DC3 covered a larger area with a potentially larger variety of
sources and chemical conditions. Although SEAC$^4$RS and DC3 both sampled the
continental US, SEAC$^4$RS had more spatial coverage and sampled more air masses at low
altitudes, while DC3 was designed to sample convective outflow air masses and had more
data at high altitudes. Although KORUS-AQ covered a much smaller area compared to
SEAC$^4$RS, KORUS-AQ data also had a large spread, which may be due to the
complicated South Korean anthropogenic sources mixed with transported air masses
(e.g., from China) and maybe biogenic sources. OA exhibits a tight correlation with
HCHO for both wildfires and agricultural fires during SEAC$^4$RS. This is because the
production of HCHO and OA is much higher in BB air masses compared to background.
This may also suggest that the emissions of OA and HCHO in these air masses are
relatively constant. SEAC$^4$RS data are chosen because it sampled fires and had state-of-
the-art, high quality measurements. More intensive fire sampling is needed to probe the
correlation between OA and HCHO across fuel types and environmental conditions.

The different slopes of OA-HCHO among different campaigns also reflect the regional or
source-driven OA-HCHO variability. Among the BB, anthropogenic and biogenic
sources, the slopes of OA vs. HCHO for BB air masses were the highest. This is
consistent with high POA emission in BB conditions (Heald et al., 2008; Lamarque et al.,
2010; Cubison et al., 2011), with low addition of mass due to SOA formation (Cubison et
al., 2011; Shrivastava et al., 2017). The slope of OA to HCHO was higher for wildfires
than agricultural fires during SEAC[4]RS though data were limited (see Table 1). This is
consistent with more OA emitted in wildfires than agricultural fires (Liu et al., 2017).
The factors driving higher OA to HCHO with wildfires are not clear and may be related
to burning conditions and fuels. For the non-BB sources, the slope of OA vs. HCHO was
highest for South Korea (KORUS-AQ), which is dominated by heavily polluted
anthropogenic sources. During KORUS-AQ, the high OA to HCHO air masses also had
high acetonitrile. By the time we sampled, most organic aerosols were secondary (Nault
et al., 2018). This indicates that the formation rates of OA and HCHO from different
emission sources contribute to the different slopes of OA-HCHO. This also indicates that
emission sources with enhanced acetonitrile tend to form more OA relative to HCHO
downwind.  The slope of OA-HCHO for California LA basin, dominated by relatively
clean anthropogenic emissions, was much lower than South Korea. The potential
difference in the anthropogenic emissions mix could contribute to the different OA-
HCHO slopes from US LA region and South Korea anthropogenic sources (Baker et al.,
2008; Na et al., 2005; Na et al., 2002). The slopes of OA vs. HCHO of SEAC[4]RS and
DC3 dominated by biogenic emissions in the SE US were in-between heavily polluted
(KORUS-AQ) and clean anthropogenic sources (CalNex). As SEAC[4]RS had the largest
geographic coverage for low altitude data over US, the campaign average slope of OA vs.
HCHO was used to represent the US region in summer. CalNex LA Basin data were used
to represent large cities as case studies.

Overall, the source dependent OA-HCHO relationships (Fig. 2) showed higher OA-
HCHO slopes of BB and heavily polluted anthropogenic sources with inefficient
combustion (e.g., KORUS-AQ) compared to biogenic and relatively clean anthropogenic
sources. This indicated that inefficient combustions contribute to the high slopes of OA-
HCHO, probably due to both enhanced primary OA and increased formation of SOA.
Enhanced pre-existing aerosols such as primary aerosols can provide more surfaces to
increase VOCs condensation and SOA formation. VOCs co-emitted from heavily
polluted anthropogenic sources can also form more SOA. It is possible to extract the
factors that govern the different OA-HCHO relationships and potentially have a universal
application of the slopes as a function of the factors (e.g., sources and combustion
efficiencies).

**3.2 Dependence on $NO_x$ and VOCs speciation**
Biogenic and anthropogenic VOCs are oxidized by atmospheric oxidants (e.g., OH as the
dominant oxidant) to form $RO_2$. HCHO is produced from the reactions of $RO_2$ with $HO_2$
or NO, with $RO_2$+NO typically producing more HCHO than $RO_2 + HO_2$ (e.g., Wolfe et
al., 2016). $RO_2$ can react with $HO_2$ or NO, or isomerize to form oxidized organic
compounds with high molecular weight and low volatility, which condense on existing
particles to form SOA. The products of $RO_2$ + NO tend to fragment instead of
functionalize and often lead to higher volatility compounds (e.g., HCHO) and thus less
SOA formation compared to the products of $RO_2$ + $HO_2$ (Kroll et al., 2006; Worton et al.,
2013). Therefore, with the same VOC, we expect more HCHO and less OA formed at
high NO conditions and vice versa. As mentioned before, $NO_2$ instead of NO is easily
measured from space and $NO_2$ typically is ~80% of $NO_x$ in the boundary layer during the
day. Therefore, $NO_2$ is used as a surrogate for the NO levels influencing OA and HCHO
production. The yields of HCHO and SOA also depend on VOC speciation (e.g., Lee et
al., 2006; Bianchi et al., 2016). Specifically, isoprene has a higher yield of HCHO than
most non-alkene VOCs (Dufour et al., 2009).

A scatter plot of OA vs. HCHO for SEAC[4]RS low altitude data is shown in Fig. 3(a). The
data are color-coded by the product of in-situ isoprene and $NO_2$, attempting to capture
time periods strongly influenced by oxidation products of isoprene at high NO
conditions. No trends are evident when the data are instead color coded by $NO_2$ or
isoprene only. This may be because isoprene (biogenic source) and $NO_2$ (anthropogenic
sources) are generally not co-located in the US (Yu et al., 2016) and isoprene is the
dominant source of HCHO compared to anthropogenic VOCs in the US (e.g., Millet et
al., 2008). This plot shows that, at high $NO_2$ and high isoprene conditions, less OA was
formed for each HCHO produced generally. The correlation coefficient of 0.45 for high
$NO_2$ and isoprene conditions during SEAC[4]RS is not very high but still shows significant
dependence of the OA-HCHO relationship on the product of $NO_2$ and isoprene,
considering that these are ambient data and other factors (e.g., different specific sources)
also play a role in determining OA-HCHO relationships. This is consistent with high NO
and isoprene conditions promote HCHO formation over SOA formation. We also looked
at the dependence on peroxy acetyl nitrate (PAN), as PAN is a product of the photo
oxidation of VOCs, including isoprene, in the presence of $NO_2$. The dependence on PAN
was not as clear as on the product of $NO_2$ and isoprene.

KORUS-AQ OA vs. HCHO, color-coded with $NO_2$, is plotted in Fig. 3(b). The OA-
HCHO ratio clearly decreased as $NO_2$ levels increased during KORUS-AQ, suggesting
that high NO conditions accelerated HCHO formation more than they did SOA
production. OA-HCHO relationships do not have dependence on local time of the day
(not shown). This further confirms that $NO_x$ is an important factor that affects the OA-
HCHO relationship.  Compared to SEAC[4]RS, the KORUS-AQ OA-HCHO ratio does not
depend on VOCs. This may be consistent with the dominant VOCs being anthropogenic
VOCs that are co-located with NO sources. This may also suggest that the anthropogenic
VOCs generally have a lower HCHO yield than does isoprene. Because OA and HCHO
were tightly correlated during CalNex and DC3, we did not parse for $NO_x$. The $NO_x$
range during DC3 low altitude data was smaller than KORUS-AQ and SEAC[4]RS. DC3
OA-HCHO relationships only had a slight dependence on $NO_2$ (not shown here), largely
due to the limited dataset. The $NO_x$ range during CalNex low altitude data was large. The
OA and HCHO correlation during CalNex was very tight and the slope of OA-HCHO did
not show clear dependence on $NO_x$, which could be due to the combination of different
VOCs sources and $NO_x$ levels.

**4. Comparison of OA-HCHO relationships: in-situ vs. GEOS-Chem**

In situ OA-HCHO relationships from SEAC[4]RS low altitude non-BB (Fig. 4a), KORUS-AQ low altitude (Fig. 4b), and SEAC[4]RS BB (Fig. 4c) air masses were compared to GEOS-Chem model simulations (Fig. 4d-4f) sampling along the corresponding flight tracks. Similar to the in situ data, GEOS-Chem model simulations also found correlations between OA and HCHO for these three regions, especially for SEAC[4]RS non-BB. GEOS-Chem was intensively validated with in situ measurements for SE US (e.g., Marais et al., 2016; Kim et al., 2015). The ratios of the slopes between OA and HCHO for the US (SEAC[4]RS), South Korea (KORUS-AQ), and wildfire cases (SEAC[4]RS) from GEOS-Chem were 1:1.1:0.4, which was different from the in situ measurements of 1:1.4:13 (Table 1). GEOS-Chem could not capture any wild fires in US during SEAC[4]RS, which is probably due to poor representation of BB emission inventory for US wildfire and also the coarse grid in GEOS-Chem. GEOS-Chem also significantly under predicted the slope of OA to HCHO for South Korea. We attribute this to a likely underprediction of anthropogenic SOA, which was dominant in South Korea, in GEOS-Chem (Schroder et al., 2018), as well as a different mix of OA and HCHO sources in the US compared to South Korea and representation of these in GEOS-Chem. Although GEOS-Chem contains isoprene chemistry with a focus on the SE US (Marais et al. 2016), there is still room to improve GEOS-Chem model especially for anthropogenic and BB sources, as well as anthropogenic OA formation mechanisms. For example, in the model biogenic sources are more important than anthropogenic sources for the OA and HCHO budgets in South Korea, which is not the case from KORUS-AQ in situ measurements. In the model, a 10% decrease of emissions from biogenic, anthropogenic and BB sources

results in a 6%, 3%, and 1% decrease in OA and 2%, 1%, and 0% decrease in HCHO
over South Korea in May 2016. However, the in situ airborne field campaign KORUS-
AQ found that OA and HCHO were higher near anthropogenic emission sources
compared to rural regions. The larger impact of biogenic sources compared to
anthropogenic sources on OA and HCHO in the model can be due to both low-biased
anthropogenic emission inventories and low-biased anthropogenic SOA. Improving
anthropogenic emissions inventories in the models can bring model results closer to
observations. Improving anthropogenic SOA, such as implementation of the SIMPLE
model, in GEOS-Chem (Hodzic and Jimenez, 2011) can also improve the model results
compared to observations. Measurements or measurement-constrained estimation with
sufficient spatial and temporal coverage can help to narrow down the key factors (e.g.,
emission inventories or chemical schemes) in GEOS-Chem to better represent VOCs and
OA globally. Furthermore, we did also find that GEOS-Chem could not capture the
observed higher slope of OA to HCHO at high altitudes (not shown), which could be due
to issues such as transport, OA lifetime, and OA production.

**5.  Relating satellite HCHO column to surface HCHO concentrations**
To utilize the derived in-situ OA-HCHO relationship, the satellite HCHO columns need
to be converted to surface HCHO concentrations. We used a vertical distribution factor $\eta$
($cm^{-1}$) (Sect. 2.5), which is defined as the ratio of surface HCHO concentrations (molec
$cm^{-3}$) to HCHO column (molec $cm^{-2}$), to estimate surface HCHO concentrations from
satellite column measurements. Zhu et al. (2017) used the same vertical distribution
factor for their study. The use of this factor is justified by the fact that the derived surface
HCHO retained the spatial pattern of the satellite HCHO column and agreed with local
surface measurements of HCHO for a multi-year average (Zhu et al., 2017).

We also investigated the main factors affecting the variation of the vertical distribution
factor $\eta$. Because the factor is determined by HCHO vertical distributions, we examined
three typical normalized HCHO vertical distribution profiles with the highest, median and
lowest $\eta$ values for the SEAC[4]RS field campaign (Fig. 5). Because the sensitivity of OA
estimates to $\eta$ was investigated with $\eta$ from different GEOS-Chem versions (Sect. 6.2),
we did not compare HCHO vertical profiles from the model to the measurements from a
comprehensive set of field campaigns. We chose SEAC[4]RS to illustrate the main factors
impacting the $\eta$ over US because SEAC[4]RS had a larger spatial coverage than DC3 and
CalNex. GEOS-Chem can generally capture the vertical profiles of measured HCHO.
Boundary layer mixing height and surface emission strength are the dominant factors in
determining the fraction of HCHO near the surface. Higher boundary layer mixing height
results in lower $\eta$ for SE US profiles, where there are biogenic sources of HCHO from
the surface and HCHO has distinct concentration difference below and above the
boundary layer. However, there are exceptions, such as for the profiles over the ocean
and the coastal regions. Although the boundary layer is shallow in these regions, a large
portion of HCHO resides above the boundary layer, resulting in low $\eta$. In these cases,
surface emissions of HCHO or precursors are very small and therefore methane oxidation
makes a large contribution to the total HCHO column. High concentrations of HCHO
(e.g., in BB plumes) lofted by convection can also impact the vertical profile (Barth et al.,
2015), which is not further investigated because OA estimates with BB influences over
US are excluded in current study. Overall, the source intensities and boundary layer
mixing height mostly determined the HCHO vertical profiles.

**6. Construction of the OA estimate**
**6.1 Variables to construct OA estimate**
As mentioned in Sect. 2.5, the OA estimate value in each grid cell was estimated from
monthly average satellite HCHO column observation by the linear Eq. (1). Satellite
monthly average HCHO column data, $\Omega_{HCHO}$, were converted to surface HCHO
concentrations by multiplying by the $\eta(i)$ factor either from climatology a priori profiles
or monthly average HCHO profiles. Surface OA was then estimated by multiplying the
derived surface HCHO concentrations with the slope $\alpha(i)$ and adding the intercept $\beta(i)$.
The slope $\alpha(i)$ and intercept $\beta(i)$ were determined from the linear regression of in situ
OA and HCHO from aircraft field campaign data. The relationship between OA and
HCHO varies but previous sections demonstrated that we can quantify the surface OA-
HCHO relationship by their regions, sources and chemical conditions (e.g., $NO_x$ and
isoprene levels). To test the impact of the chosen OA-HCHO relationship on the
calculated OA estimate, the OA estimate in the US was calculated using four different
methods (see Table 2). The OA estimate was calculated on the monthly time scale,
largely because OA estimate is based on OMI HCHO observations and uncertainty
weighted average for a time scale of about one month (Gonzalo et al., 2015; Zhu et al.,
2016) is needed to reduce the noise in daily OMI HCHO data. With improved satellite
HCHO data from TROPOMI, higher time resolution (e.g., weekly average) HCHO data
could be useful to estimate OA in the future.

**6.2 OA estimate over US**


The monthly average surface OA estimates over the US in August 2013 using SEAC⁴RS
lump-sum slope and intercept (see Table 2) with different $\eta$ are shown in Fig. 6a and 6b.
Because BB regions in the US are not covered by smoke continuously during a period of
time and it is challenging for satellite retrieval to separate thick BB plumes and clouds
without information on the time and location of the burning, thick BB events (OMI UV
Aerosol Index (UVAI) > 1.6) (Torres et al., 2007) were excluded and shown as the blank
(white) grid cells in Fig. 6a and 6b. The same filter was also applied to aerosol extinction
and GEOS-Chem OA abundance. To evaluate the representative quality of the OA
estimate, OA estimate data were compared to the EPA IMPROVE ground sites corrected-
OA measurements over the US and SEARCH ground sites OA measurements in the SE
US (Sect. 2.2). The locations of IMPROVE and SEARCH sites are displayed in Fig. 6e
as small and large dots, respectively. The dot color represents the average OA mass
concentrations for August 2013.

Considering the uncertainties in satellite HCHO measurements, in using the campaign
lump-sum OA-HCHO relationship to represent spatial resolved OA, in HCHO vertical
profiles, and in ground IMPROVE network measurements, the correlation (correlation
coefficient r = 0.56) between the OA estimate and corrected IMPROVE network
measurements (Fig. 6f and 6g) is reasonably good and indicates that the OA estimate can
generally capture the variation of OA loading over the US. First, the correlation
coefficient between HCHO SAO retrievals and in situ measurements during SEAC⁴RS
was not high  (r = 0.24) but this may be partly because they were not sampled at the same
time. The uncertainty in HCHO SAO data was likely less than 76%. Second, the
uncertainty in applying a campaign lump-sum OA-HCHO relationship to individual
spatial resolved satellite HCHO data to estimate OA induced an uncertainty of 41%
according to the correlation coefficient of OA-HCHO in the field campaign. Third, $\eta$ in
the Fig. 6a OA estimate was from GEOS-Chem v9-02 output for the specific month
August, 2013.  $\eta$ in the Fig. 6b OA estimate was from GEOS-Chem v9-01-03
climatology, the same as satellite data a priori profiles. The good correlations of OA
estimates with IMPROVE OA indicate that OA estimates are not very sensitive to $\eta$ from
different model versions. The largest difference between the two OA estimates is their
concentrations over East Texas. There are no IMPROVE OA measurements in the East
Texas to evaluate which works better. Fourth, the uncertainties in IMPROVE OA
measurements, such as using a constant correction factor to correct the partial
evaporation across all SE US sites, and the spatially dependent OA/OC ratio (Tsigaridis
et al., 2014), may also have contributed to the discrepancies between the OA estimate and
EPA IMPROVE sites OA. Therefore, higher quality of satellite HCHO data and refining
OA-HCHO relationships will help improve our OA estimate products. These combined
with a spatially resolved IMPROVE OA correction factor and OA/OC ratios will help
improve the correlation coefficients between OA estimates and IMPROVE OA.

The linear correlation between the OA estimate and IMPROVE OA measurements
yielded a slope of 0.62 or 0.60, indicating that the OA estimate underestimated OA. First,
the different data collection time for satellite data, in situ measurements and ground
observations could contribute to the bias. Satellite HCHO data were measured in mid-
day, in situ airborne OA and HCHO were measured during the daytime and IMPROVE
network organic carbon was collected day and night. Because ground OA in the SE US
were observed to have little diurnal variation (Xu et al., 2015; Hu et al., 2015), the
different sampling time of ground and airborne OA probably does not have a significant
impact on the comparison of OA estimate and IMPROVE OA. Surface HCHO has
evident diurnal profiles with the highest concentrations around the mid-day (Kaiser et al.,
2016), which could add uncertainties to OA estimate when using inconsistent time ranges
of satellite HCHO data measured in the mid-day and in situ airborne OA-HCHO
relationships measured in the daytime. The SEAC$^4$RS HCHO concentrations were
converted to 1:30 pm concentrations according to the average HCHO diurnal profile from
the Southern Oxidant and Aerosol Study (SOAS) (Kaiser et al., 2016). The OA-HCHO
relationship with HCHO converted to 1:30 pm yielded a slope of 5% lower than the
original OA-HCHO relationship. Second, the potential uncertainty ($\pm$30%) in OA/OC
ratio could also contribute to the systematic difference because we used OA/OC of 2.1
and studies (e.g., Pye et al., 2017; Canagaratna et al., 2015) showed that the OA/OC can
range from 1.4 to 2.8.   Third, the potential underestimation of HCHO from satellite
retrieval (by −37%) (Zhu et al., 2016) compared to SEAC$^4$RS may be one of the most
important reasons that cause the systematic difference (low slope) between the OA
estimate and IMPROVE OA according to Eq. (1). Satellite HCHO data corrected by the
low bias (by −37%) (Zhu et al., 2016) will increase our slopes of 0.60-0.62 to be close to
the unity.

SEARCH OA data were also used to compare to the OA estimate. The correlation was
good for August 2013. Although the SEARCH network OA measurements have better
accuracy, the number of SEARCH sites is limited (5 sites). The correlation of OA
estimate and SEARCH OA varied dramatically 2008-2013 (Fig. S1). GEOS-Chem OA
did not correlate with SEARCH OA except for the year 2013 (Fig. S1). As the
IMPROVE network has more sites and spatial coverage, we used IMPROVE network
data as ground OA measurements for comparison in the remainder of the discussion.

**6.3 Comparison to aerosol extinction from AOD**
To further evaluate the method of using satellite HCHO to derive an OA surface estimate,
satellite aerosol measurements were used to approximate surface OA extinction for
comparison. Satellite measurements of AOD were converted to surface extinction (see
Sec. 2.6). Studies showed that OA were a dominant component of aerosol mass and
extinction during SEAC$^4$RS (Kim et al., 2015; Wagner et al., 2015) and the fractions of
OA were relatively constant (interdecile 62-74%) (Wagner et al., 2015). Therefore AOD
variation is expected to generally reflect the OA variation during SEAC$^4$RS. Satellite
measurements from MISR can provide more aerosol property information to apportion
total AOD to AOD of a subset of aerosols with small to medium size and round shape,
which can better capture OA, when AOD is above 0.15 to 0.2 (Kahn and Gaitley, 2015;
personal communication with R. Kahn, 2018). Because MISR cannot distinguish OA and
other submicron aerosol components (e.g., sulfate and nitrate) and would cut off low
AOD data which accounted for near half of the data over US, we used total AOD to
derive extinction for our comparison. The AOD-derived extinction map is shown in Fig.
6(c), and the scatter plot of AOD-derived extinction and EPA corrected OA is displayed
in Fig. 6(h). The same filter of high AI was also applied to AOD-derived extinction to
remove BB plumes. Generally, the derived aerosol extinction had a correlation with
IMPROVE OA, but the correlation was not as good as for the OA estimate with
IMPROVE OA. The high surface aerosol extinctions (> 150 $Mm^{-1}$) (outliners in the
scatter plot) were located in the SE US and therefore were not due to potential
contribution of dust and nitrate altering the shape of vertical profiles outside of the SE
US. This indicates that the OA estimate derived from HCHO may be better than AOD at
representing the concentrations of OA, even for the regions where AOD is dominated by
OA (Xu et al., 2015).

**6.4 Comparison to GEOS-Chem OA**
Surface OA over the US from a GEOS-Chem simulation for August 2013 is shown in
Fig. 6(d), and the scatter plot of GEOS-Chem OA with IMPROVE OA is in Fig. 6(i).
Although HCHO vertical profiles from GEOS-Chem were used in OA estimate, the
GEOS-Chem simulation had a coarser resolution than OA estimate. To be comparable to
the OA estimate, the scatter plot Fig. 6(i) used GEOS-Chem results for the grid squares
that overlaped with individual IMPROVE sites. Compared to the OA estimate, GEOS-
Chem OA had a similar correlation coefficient with IMPROVE OA. Although the
GEOS-Chem OA plot appeared more scattered, there were many GEOS-Chem data
points close to zero when IMPROVE OA was low, making the overall correlation
coefficient similar to that for the OA estimate. GEOS-Chem under predicted IMPROVE
OA more with a slope of 0.57 compared to the OA estimate. This is consistent with
underprediction of anthropogenic OA in Marais et al. (2016).

**6.5 OA estimate with different OA-HCHO relationships**
OA were estimated with different OA-HCHO relationships for 4 cases (Table 2). LUMP-
SUM was using the non-BB SEAC$^4$RS campaign lump-sum relationship, the same as
shown in Fig. 6; ISOP-NOx was using non-BB SEAC$^4$RS $NO_2$ and isoprene dependent
relationship; URBAN was using CalNex for large urban cities and SEAC$^4$RS lump-sum
for other US regions; and COMBINE was using CalNex for large urban cities and $NO_2$
and isoprene dependent non-BB SEAC$^4$RS for other US regions. The OA estimates from
the 4 cases (Table 2) were compared to IMPROVE OA and the correlation coefficients
are shown in Fig. 7. In general, OA estimate results from the four cases were similar.

The details about how to implement chemical factors dependent OA estimates for the
four cases are also provided in Table 2. Including the $NO_2$–isoprene-dependent OA-
HCHO relationship (ISOP-NOx case) showed a similar (or slightly worse) correlation
between the OA estimate and IMPROVE OA. OMI $NO_2$ column observations were used
to represent surface $NO_2$ levels and surface isoprene emissions from MEGAN were used
to represent surface isoprene concentrations, assuming that $NO_2$ column observations
reflect surface $NO_2$ distributions and isoprene emissions reflect the concentrations of
isoprene due to its short lifetime (~1 hr). The detailed implementation is provided in the
notes in Table 2. As the in situ data showed a moderate $NO_2$-isoprene-dependent OA-
HCHO relationship, we attributed this to the locations of IMPROVE site at rural regions,
the uncertainty in IMPROVE network measurements, the uncertainty in isoprene
emissions from MEGAN, or factors (e.g., source-dependent OA-HCHO) that also need to
be taken into account when determining the specific OA-HCHO relationship. Satellite
OMI $NO_2$ data (at 1 : 30 pm) were used to represent $NO_2$ levels, big cities were defined
as $NO_2 > 4 \times 10^{15}$ molec $cm^{-2}$, and the CalNex in situ OA-HCHO relationship was
applied for big cities. It turned out that only 1 IMPROVE site (San Gabriel, SAGA1) near
LA was affected by high $NO_2$ and led to the insignificant change in URBAN compared to
LUMP-SUM. This is not unexpected because IMPROVE sites are in rural regions. The
OA estimate in SAGA1 decreased from 1.88 μg $m^{-3}$ from LUMP-SUM to 0.17 μg $m^{-3}$ in
URBAN while the measured OA in IMPROVE SAGA1 was 1.52 μg $m^{-3}$. This may infer
that CalNex is not very consistent with $SEAC^4RS$ due to different sampling instruments,
strategies and seasons. Lowing the $NO_2$ threshold when defining big cities did not help
improve the agreement either.

Because separating large urban areas and other regions and applying a simple chemical
regime dependent in situ OA-HCHO relationship did not improve the agreement between
the OA estimate and IMPROVE OA, we used the lump-sum OA-HCHO relationship to
derive the OA estimate (shown in Fig. 6). $SEAC^4RS$ and DC3 only had a few low altitude
data in the Midwest and did not cover the Northeast US. The measured OA-HCHO
relationship in the Midwest did not show significant difference from the SE US. The
scatter plots (Fig. 6f and 6g) of OA estimates and IMPROVE OA do not show outliners
for the Northeast and Midwest. This indicates that using the $SEAC^4RS$ lump-sum OA-
HCHO relationship can reasonably capture regions outside of the SE US.

**6.6 Temporal variation of the agreement between OA estimate and IMPROVE OA**

Besides August 2013 (see Fig. 6), the correlations between the OA estimate and

IMPROVE OA for the summer months June-July-August 2008-2013 were also examined

and shown in Fig. 7. Generally, the correlation coefficients between the OA estimate and

IMPROVE OA were >0.5 for summer months of the years investigated. The correlation

coefficients were generally higher in June compared to July and August. The lower

average temperature in June might be related to the higher correlation coefficients.

IMPROVE network aerosol samples were transported at ambient temperature in a truck

and more organic vapors likely evaporated at higher temperature. The different

temperatures and distances from IMPROVE sites to the laboratory may lead to

inhomogeneous evaporation among the samples and result in lower correlation

coefficients. Although higher temperatures in July and August may also lead to more BB,

average aerosol index over the US was not higher in July (mean: 0.35) and August

(mean: 0.36) compared to June (mean: 0.39) for these years. The underlying cause for the

lowest correlation coefficients in July and August 2012 is not clear and may be related to

the severe drought in 2012 (Seco et al., 2015). The correlation coefficients were also low

for the linear regressions (not shown) of IMPROVE OA with both GEOS-Chem OA and

AOD-derived extinction. Because the lowest correlation coefficients were consistently

observed for multiple OA-related products and not just the OA estimate, we attributed

this to uncertainties in the IMPROVE OA measurements or some unknown bias shared

by the satellite HCHO, GEOS-Chem OA, and satellite AOD.

728

**6.7 South Korea OA estimate**

We attempted to estimate an OA estimate for South Korea, using airborne in situ measurements of OA and HCHO from the KORUS-AQ field campaign (Aknan and Chen, 2017) and SAO OMI HCHO measurements. The National Institute of Environmental Research (NIER) ground sites OC measurements during KORUS-AQ over South Korea could be used to validate the OA estimate. However, OMI HCHO measurements were below the detection limit (Zhu et al., 2016) in May 2016. Also, there were no OMI data available in June 2016 when airborne measurements and ground sites OC measurements were available during KORUS-AQ. Because an OA estimate for South Korea could not be well retrieved and validated, it was not presented in this study. Although an OA estimate for South Korea could not be retrieved in the current study, the consistency in the dependence of OA-HCHO relationships on chemical factors (e.g., emission sources, $NO_x$, and altitudes) provides important information for potential application of chemical factors dependent OA-HCHO relationships to the geographical domain beyond the continental US, especially with improved satellite HCHO data from Tropospheric Monitoring Instrument (TROPOMI).

**7   Limitations of the OA estimate and future work**

Because the OA estimate is based on satellite HCHO data, the detection limit of satellite HCHO data affects the quality of the OA estimate. Currently, due to the limited sensitivity of OMI for HCHO, the OA estimate is valid only when high levels of HCHO are present, such as during summer time and near large HCHO sources. With the new TROPOMI satellite instrument and future missions TEMPO and GEMS, satellite HCHO

measurements will have higher spatial and temporal resolutions and lower detection
limits. These higher quality satellite HCHO measurements will improve the quality and
spatial and temporal coverage of our OA estimate.

Because the OA estimate uses the relationship of in situ HCHO and OA measurements,
the coverage of in situ aircraft field campaigns will impact the OA estimate quality.
Currently, in situ airborne measurements of OA and HCHO focus on the continental US.
Extending measurements to regions such as Africa BB, South America, and East Asia,
where HCHO and OA have high concentrations, will increase the spatial coverage of the
OA estimate product. Ground site measurements of OA with consistent quality control in
those regions will also be important for validating the OA estimate.

Improvement of satellite HCHO retrieval during the BB cases will also improve OA
estimate quality. BB cases with high UV aerosol index over the US were excluded in the
current OA estimate. With improvement in the satellite retrieval of HCHO, we may be
able to estimate OA during BB cases over the US. Upcoming field campaigns such as the
Fire Influence on Regional and Global Environments Experiment - Air Quality (FIREX-
AQ) will provide opportunities to improve the OA estimate in BB cases in the US.

This OA estimate method has limitations in remote regions far away from HCHO
sources. Because the lifetimes of HCHO (1-3 hours) and OA (1 week) are different, the
slopes and intercepts between HCHO and OA are expected to change when air masses
are aged (e.g., in remote regions). HCHO is close to being in steady-state with production
rates roughly equal to loss rates while OA is not in steady-state with a lifetime of a week.
Therefore, OA can be accumulated relative to HCHO when air masses are aged. OA vs.
HCHO from SEAC[4]RS and KORUS-AQ field campaigns, color-coded with altitude as an
indicator of air mass age, are plotted in Fig. S2 (a) and (b), respectively. A relative
depletion of HCHO at high altitudes was observed due to its shorter lifetime. This also
suggests that, at remote regions far away from the sources, the ratios of OA and HCHO
could be much higher and the relationship between OA and HCHO derived near the
sources may no longer apply. On the other hand, the lifetime of 1-3 hrs for HCHO does
not imply that the OA estimate only work within this timescale. HCHO is formed from
oxidation of transported gas phase VOCs, including the oxidation products of the primary
emitted VOCs, as well as of the slower reacting VOCs (e.g., Ethane and Benzene). Most
gas-to-particle oxidation processes that might produce HCHO can last up to 1-2 days
(Palm et al., 2018). Fig. S3 shows the ratios of OA and HCHO did not change
significantly downwind for the Rim Fire plume for about 1 day of aging, which was
determined by the distance from the source and the wind speed. A lower photolysis rate
of HCHO in the plume can also contribute to this. However, we do not expect the
relationship of OA and HCHO to remain past 1-2 boundary layer ventilation cycles (Palm
et al., 2018). Although OA-HCHO relationships depend on air mass age, it does not
largely affect our study for monthly average surface OA over continental US because our
OA estimates showed reasonably good agreement with ground sites IMPROVE OA
measurements. This also indicates that SOA are enhanced near the source regions
statistically. Nault et al. (2018) also showed the production of HCHO and SOA are
similar and plateau around 0.5 − 1 photochemical day. So, in the near field of emissions
and chemistry, the productions of these two species are similar; however, outside of near
field of emissions and rapid chemistry, the long lifetime of OA vs the steady state of
HCHO would start controlling the slopes and correlations.

**8   Summary**
We have developed a satellite-based estimate of the surface OA concentration ("OA
estimate") based on in situ observations. This estimate is based on the empirical
relationships of in-situ OA and HCHO for several regions. OA and HCHO share VOC
sources with different yields and lifetimes. Using surface OA and HCHO linear
regression slopes and intercepts we can relate surface HCHO to OA. To estimate the
surface HCHO concentration from satellite HCHO column, we used a vertical
distribution factor $\eta$ from either climatology satellite data a priori profiles or updated
model run for specific period, which is largely determined by boundary layer height and
surface emissions and found to reasonably retrieve surface HCHO from column HCHO.

The OA estimate over the continental US generally correlated well with EPA IMPROVE
network OA measurements corrected for partial evaporation, with a biased low slope of
0.62 or 0.60, mostly due to underestimation of HCHO concentrations from the OMI
HCHO retrieval. The good correlations are not only for the time during SEAC[4]RS but
also for most summer months over several years (2008-2013) investigated. Compared to
aerosol extinction derived from AOD, the OA estimate had slightly higher correlation
coefficients with IMPROVE OA. GEOS-Chem can predict OA with a similar correlation
coefficient with IMPROVE OA compared to the OA estimate when GEOS-Chem was
intensively validated with in situ measurements for SE US. Better satellite HCHO data
from TROPOMI and future TEMPO and GEMS and extending spatiotemporal coverage
of in situ measurements will improve the quality and coverage of the OA estimate.

**Author Contribution:**
JL performed the analysis and wrote the paper. TFH directed the research topic and
discussed the analysis with JL. TFH, GMW, JSC, AF and CW provided in situ HCHO
measurements. JLJ, PCJ, and BAN provided in situ OA measurements. EAM provided
GEOS-Chem model results. GGA and KC provided satellite HCHO data. HTJ provided
MODIS AOD data. TBR provided in situ $NO_2$ measurements. AW provided in situ
isoprene and acetonitrile measurements. GMW, TFH, JSC, JLJ, BAN, PCJ, EAM, and
GGA provided constructive comments to help improve the paper. All authors had
reviewed and edited the paper.

**Acknowledgements:**
JL, TFH, GMW, and JSC were supported by NASA grant NNH15ZDA001N and
NNH10ZDA001N. BAN, PCJ, and JLJ were supported by NASA grant NNX15AT96G
and 80NSSC18K0630. AW and PTR-MS measurements during DC3, SEAC[4]RS and
KORUS-AQ were supported by the Austrian Federal Ministry for Transport, Innovation
and Technology (bmvit) through the Austrian Space Applications Programme (ASAP) of
the Austrian Research Promotion Agency (FFG). The PTR-MS instrument team (P.
Eichler, L. Kaser, T. Mikoviny, M. Müller) is acknowledged for their field support. We
thank E. Edgerton for providing the SEARCH network data.

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

Tables
Table 1. Linear regression parameters for OA vs. HCHO at low altitudes (<1 km)

| | US (SEAC[4]RS) | US (DC3) | US (CalNex) | South Korea (KORUS-AQ) | Wild Fires (SEAC[4]RS) | Agricultural Fires (SEAC[4]RS) | SEAC[4]RS Low NO$_2$ and Isoprene | SEAC[4]RS high NO$_2$ and Isoprene |
|---|---|---|---|---|---|---|---|---|
| **In situ measurements OA v.s. HCHO** | | | | | | | | |
| Slope [a] | 1.93±0.07 | 1.30±0.10 | 1.34±0.02 | 2.75±0.05 | 25.08±0.30 | 3.22±0.37 | 2.39±0.09 | 1.45±0.19 |
| Slope [b] (×10$^{-11}$) | 9.61±0.34 | 6.49±0.49 | 6.66±0.09 | 13.7 1±0.25 | 125.05±1.49 | 16.04±1.85 | 11.9±043 | 7.25±0.96 |
| Intercept [c] | 0.34±032 | 1.10±0.30 | -0.90±0.06 | 1.36±0.22 | −6.85±2.80 | 10.41±5.82 | −1.14±0.37 | 1.14±1.22 |
| Correlation coefficient r | 0.59 | 0.76 | 0.88 | 0.70 | 0.97 | 0.85 | 0.64 | 0.45 |
| Number of points (1 min avg) | 1506 | 134 | 1772 | 3425 | 515 | 32 | 1138 | 226 |
| **GEOS- Chem model sampled along the flight track OA v.s. HCHO** | | | | | | | | |
| Slope [a] | 1.25±0.03 | | | 1.39±0.05 | 0.48±0.05 | | | |
| Slope (×10$^{-11}$) | 6.21±0.14 | | | 6.95±0.23 | 2.37±0.22 | | | |
| Intercept | −1.32 ±0.11 | | | 1.88 ±0.07 | 0.12±0.03 | | | |
| Correlation Coefficient r | 0.76 | | | 0.43 | 0.53 | | | |

[a] The unit of the slope is g g$^{-1}$.
[b] The unit of the slope is pg molec$^{-1}$.
[c] The unit of the intercept is μg m$^{-3}$.
The uncertainties are one standard deviation.

Table 2. Cases to estimate OA surface concentrations, based on the choice of slope and
intercept from a linear regression relationship between OA and HCHO data found in
Table 1.

| LUMP-SUM[a] | Using non-BB SEAC[4]RS relationship to represent all continental US |
|---|---|
| ISOP-NOx[b] | Using NO$_2$ and isoprene dependent non-BB SEAC[4]RS relationship for all continental US |
| URBAN | Using the CalNex LA Basin relationship for large urban cites and the non-biomass burning SEAC[4]RS relationship for other US regions |
| COMBINE[b] | Using the CalNex LA Basin relationship for large urban cites and the NO$_2$ and isoprene dependent non-BB SEAC[4]RS relationship for other US regions |


[a]SEAC[4]RS was chosen to represent all continental US because it had the largest horizontal and vertical
coverage.
[b] In cases ISOP-NO$_x$ and COMBINE, when the product of NO$_2$ column (Sect. 2.3) and surface isoprene
emission rate (Sect. 2.4) was above threshold of $5 \times 10^{27}$ molec cm$^{-2}$ atom C cm$^{-2}$ s$^{-1}$, the slope and intercept
from SEAC[4]RS high isoprene and NO$_2$ conditions were used. When the NO$_2$ column−isoprene emission
product was below that threshold, the slope and intercept from SEAC$^4$RS low isoprene and NO$_2$ conditions
were used. Threshold of "Isoprene × NO$_2$" was determined by its mean value over SE US (83° - 96° W and
32° - 35°N). Large urban cities were categorized with high NO$_2$ vertical columns ($>4 \times 10^{15}$ molec cm$^{-2}$)
(Tong et al., 2015) based on the satellite NO$_2$ levels over LA.  Isoprene emissions instead of concentrations
were used because global models use isoprene emission inventory to simulate isoprene concentrations and
isoprene emission inventory is easier to access. Since isoprene has a short-lifetime of up to a few hours
(Guenther et al., 2006), the emissions have a similar spatiotemporal distribution as the concentrations.


Figures

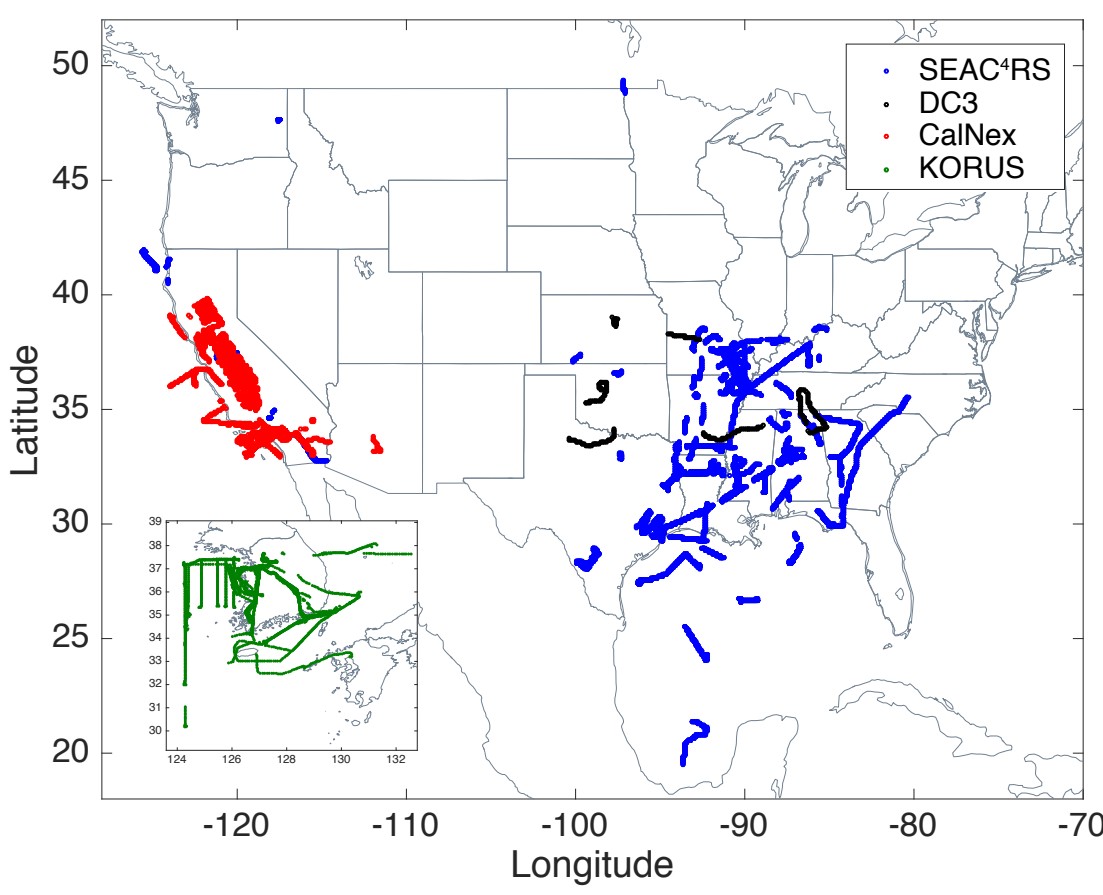

Figure 1. Flight tracks of airborne field campaigns SEAC$^4$RS (blue), DC3 (black),
CalNex (red) and KORUS-AQ (green) with altitudes (< 1 km), of which in situ OA and
HCHO measurements were used.

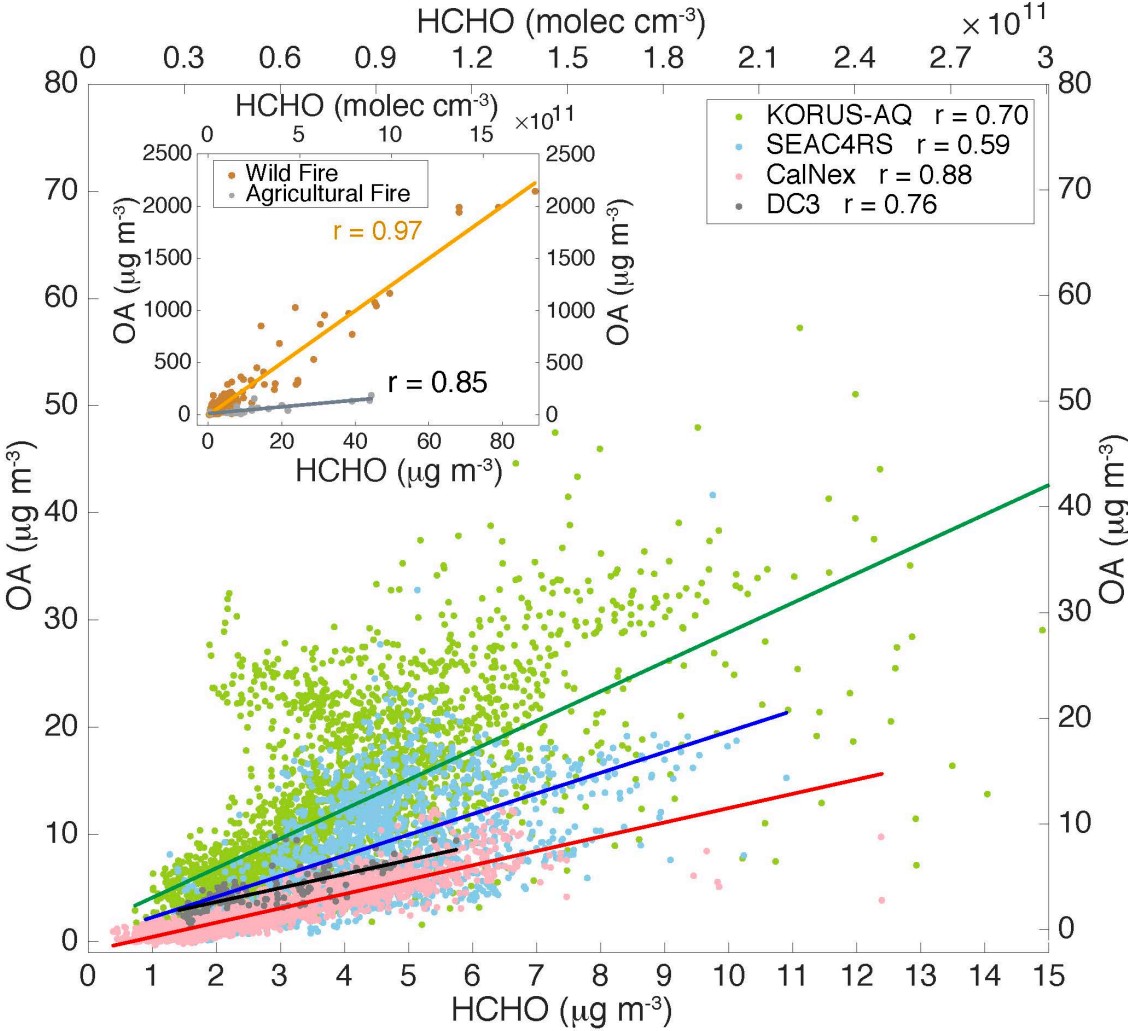

Figure 2
Scatter plots of in situ OA (µg m$^{-3}$) vs. HCHO (µg m$^{-3}$ or molec cm$^{-3}$) from SEAC$^4$RS
(excluding biomass burning) (blue), DC3 (dark grey), CalNex (pink), and KORUS-AQ
(green) low altitude (< 1 km) data. Inset shows wildfire (brown), and agricultural fire
(grey) SEAC$^4$RS data. SEAC$^4$RS biomass burning cases are defined as acetonitrile > 200
pptv. The linear regression fits are shown as the darker lines and correlation coefficients
are provided.

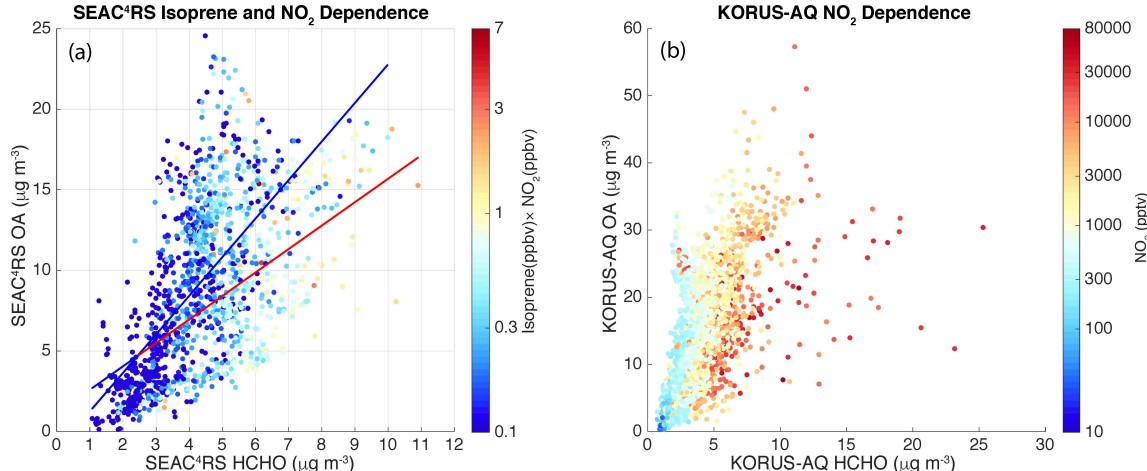

Figure 3. (a) A scatter plot of OA vs. HCHO for SEAC⁴RS non-biomass burning low
altitude data color-coded with the product of $NO_2$ and isoprene in log scale. The red and
blue lines are the linear regression fits of high (> 0.5) and low (<0.5) product of $NO_2$
(ppbv) and isoprene (ppbv), respectively. (b) A scatter plot of OA vs. HCHO for
KORUS-AQ data color-coded by log($NO_2$).

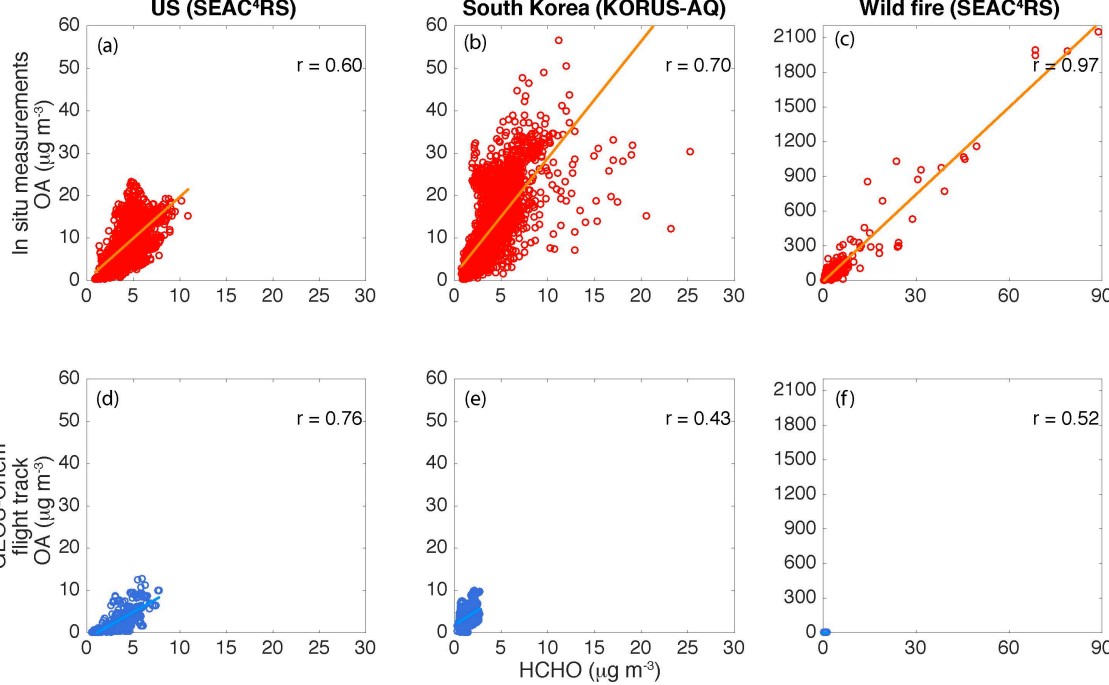

Figure 4 Scatter plots of OA vs. HCHO for US (SEAC⁴RS altitude < 1 km non-biomass
burning), South Korea (KORUS-AQ altitude < 1 km) and wildfire (SEAC⁴RS) from in
situ measurements (a, b, c) and GEOS-Chem outputs sampled along the flight tracks
(d,e,f).

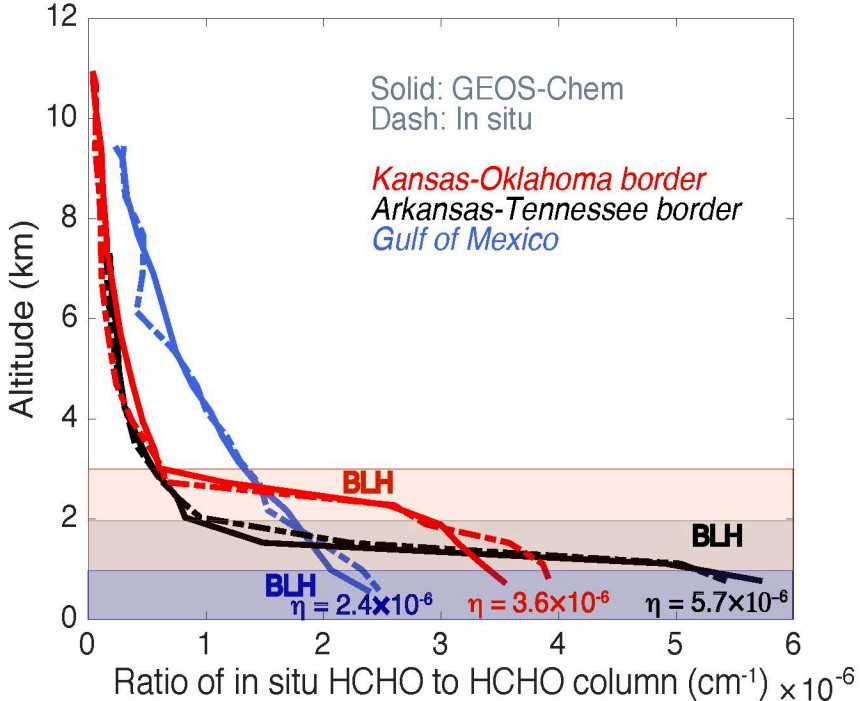

Figure 5. Three typical vertical profiles of the ratio of in situ HCHO concentrations
(molec cm$^{-3}$) to integrated HCHO column from SEAC$^4$RS flight track. These three
profiles were located at Kansas-Oklahoma border (red), Arkansas-Tennessee border
(black), and Gulf of Mexico (blue). Solid curves were from GEOS-Chem results and the
dashed were from ISAF measurements. HCHO columns were integrated HCHO
concentrations of these vertical profiles extrapolated from 0 to 10 km, assuming the
HCHO below and above the measured HCHO vertical profiles were the same as the
HCHO at the lowest and highest altitudes sampled, respectively. The boundary layer
heights (BLH) of these three profiles are plotted by the shaded areas.

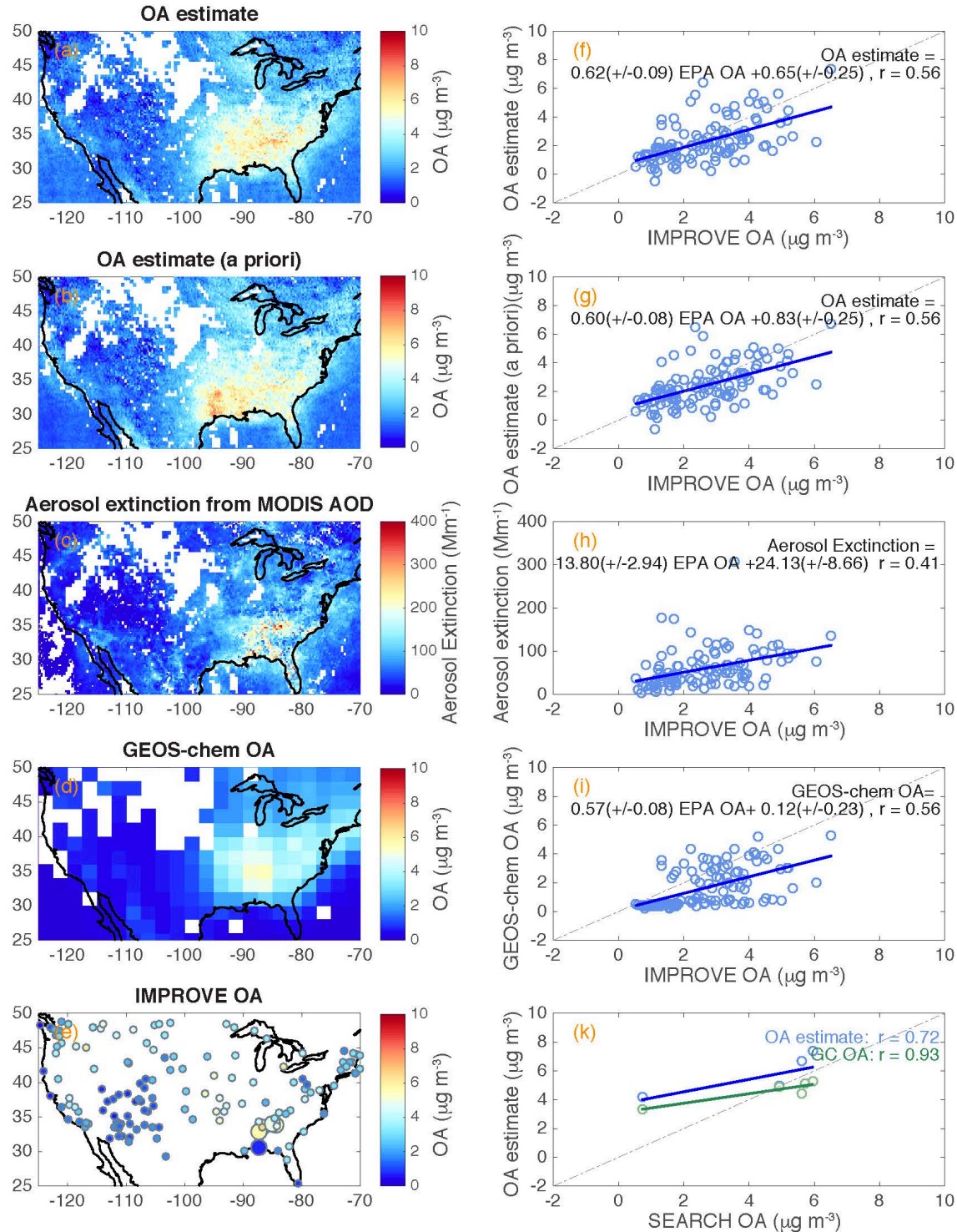

Figure 6. (a) The maps of (a) surface OA estimate (Case 1) with η from GEOS-Chem v9-
02, (b) surface OA estimate (Case 1) with η from a priori profiles, (c) surface aerosol
extinction derived from MODIS AOD, (d) GEOS-Chem simulated surface OA, and (e)
EPA IMPROVE (small dots) and SEARCH (large dots) network ground sites color coded
with OA concentrations for August 2013. The scatter plots of (f,g) surface OA estimate,
(h) surface aerosol extinction derived from MODIS AOD, and (i) surface GEOS-Chem
OA vs. EPA IMPROVE network ground sites OA. IMPROVE sites OA were corrected
for evaporation. (k) The scatter plots of surface OA estimate and GEOS-Chem OA vs.
SEARCH network ground sites OA for August 2013. GEOS-Chem OA and OA estimate
did not have good correlations with SEARCH OA for other years (SI). For the scatter
plots, linear regressions are shown (blue and green lines) and regression equations and
correlation coefficients for the scatter plots are listed. The dashed lines in the scatter plots
indicate the 1 : 1 line. Biomass burning data (UV aerosol index > 1.6) were excluded in
all panels.

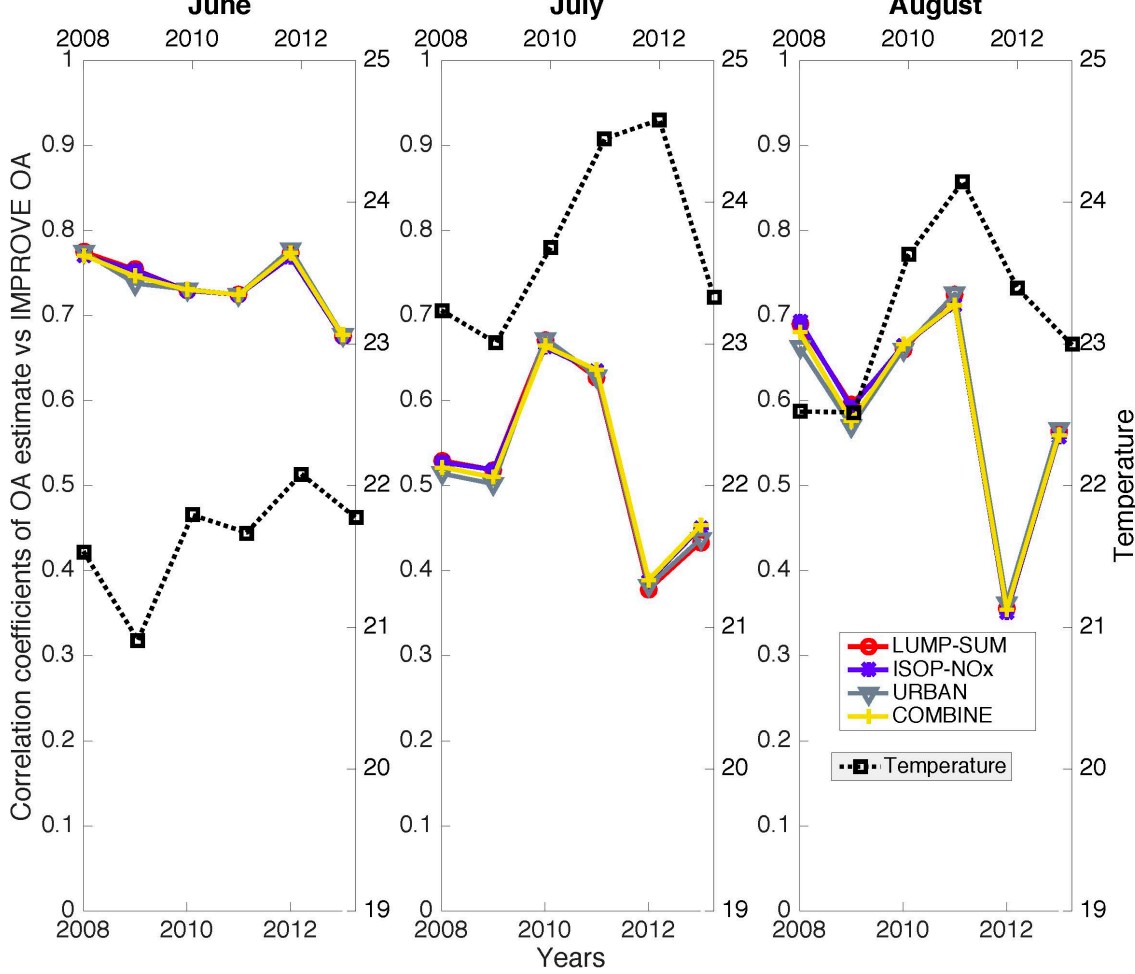

Figure 7. The correlation coefficients of the linear regression between the OA estimate
from 4 case (red, blue, gray, and yellow) vs. EPA-corrected OA from 2008 – 2013 for
June, July, and August. The monthly average ambient temperature is in black.