# Peer review of "organic aerosol abundance"

_Atmospheric Chemistry and Physics, 2018_

## Referee Comment (RC1) · Anonymous Referee #1 · 13 Sep 2018

This study aims to use the information in HCHO satellite observations, along with a model estimate of the vertical distribution, and in situ HCHO:OA relationships, to estimate surface OA concentrations over the United States. This is an interesting concept, though it is challenged with many uncertainties and assumptions and the manuscript is a bit unfocused. Here are the major issues that I have identified:

1. The title is vague and somewhat inaccurate. The estimate involves satellite, in situ observations AND a model, the "hybrid" should be described as such. Given that the results presented are US only, "over the continental United States" should be specified in the title. And finally, I would recommend that the authors modify the title to include "formaldehyde" so that it is clear that this is not an estimate based off of satellite AOD.

2. What are the implications of the statement on lines 88-89 that "isoprene SOA is

not the dominant source of SOA in summer" for the GEOS-Chem simulation in that region? The study refers several times to the GEOS-chem simulation for the SEUS being "extensively validated" (lines 52, 87, 239) but seems to refer to isoprene SOA. Does the simulation capture the non-isoprene components of SOA as well?

3. The fire analysis (inset Figure 2) is not very convincing given the small number of points. Why did the authors not include data influenced by fire from other regions in their analysis? In particular, the agricultural fire analysis appears to rely on very few points and the discussion of those results in lines 324-326 and 344-346 should be tempered.

4. The relationship between OA and HCHO seems to break down as you increase NOx (it's not all that strong to begin with, see point 7). The correlation coefficient of 0.44 implies under high NO2 less than 20% of the variability of OA can be explained by HCHO. An OA estimate based off of this weak relationship does not seem credible. Also: was NO2 not measured during CalNex and DC3 (Lines 176-183)? Why are these campaigns not parsed for NOx?

5. A key part of this analysis is the conversion of HCHO column to surface using the GEOS-Chem model. Are the AMFs used in the OMI product consistent with the eta's used here? How does the model treat boundary layer mixing? Are the results sensitive to this? Why do the authors only show SEAC4RS profiles of HCHO? How does the model perform for DC3 and CalNex? Lines 455-456 discusses the potential for BB plumes to impact the vertical profile; this could be investigated with the DC3 data.

6. The South Korea analysis is a null result and adds little to the paper. I suggest that the authors eliminate Section 6.7 and earlier analysis of KORUS-AQ data. They could re-cap in the Conclusions that a similar analysis was attempted for South Korea but was precluded by low HCHO from OMI.

7. Need to develop a more detailed and quantitative discussion of uncertainties. Several potentially large uncertainties are discussed on lines 501-508 and Section 7. The

authors should strengthen and consolidate this discussion to outline all the assumptions and potential impact on their analysis. In particular, I note that the correlations between HCHO and OA in the in situ observations are actually not all that high to begin with (in the case of SEAC4RS HCHO only explains 35% of the variability in OA). Also many assumptions are based solely on the SEUS:

a. That OA accounts for a large fraction of submicron aerosol (lines 277-278). What about in other regions of the US?

b. That the vertical profile of OA matches the total aerosol (line 284) – this has not been shown outside of SEUS and the potential for dust, nitrate, smoke plumes to alter this should be discussed

c. Applying the OA:HCHO relationship from SEAC4RS for the entire US. The authors tested using data from the LA Basin as well, but how would OA (and HCHO) differ in the Midwest and NEUS? In Section 6.2 the authors discuss the ability of the OA estimate to capture IMPROVE observations – how good is the estimate outside of the SEUS, where observations were used to construct the HCHO:OA relationship?

MINOR

1. The abstract is missing a few key descriptors: that the analysis is performed for summertime only, that the estimate of near-surface OA is for the United States, and that that estimate also relies on the GEOS-Chem vertical profile of HCHO.

2. Lines 155-166: how do ISAF and DFGAS measurements compare during SEAC4RS?

3. Line 174: specify what temperature was assumed for STP (273 and 298K are both used)

4. Line 180: correct "for daytime NO"

5. Section 2.3: The spatial resolution of the products and and details of gridding and

averaging are missing.

6. Lines 273-277: This sentence is awkwardly phrased ("can estimate the fraction of OA" and then the sentence says it can't do this...). Also why is MISR discussed here and later (lines 525-531) when it is not used in this study?

7. Line 280: need to define Aext in the text

8. Line 294: insert "near-surface in situ"

9. Line 298: define "BB" in text

10. Line 402/Section 2.4: The emissions are not described in the model description section.

11. Section 6.1: Are daily HCHO columns converted to surface concentrations using daily etas and then averaged to monthly values, or is the analysis performed using monthly means for everything?

12. Line 488-9: A correlation coefficient of 0.56 is pretty modest. When only 31% of the variation is captured it is not accurate to state that the estimate "generally captured the variation" of the observations.

13. Lines 537-538: The authors state their goal in using the extinction data here at the end of the Section. I suggest that the authors explain more clearly at the start of Section 6.3 why they are exploring extinction.

---

## Referee Comment (RC2) · Anonymous Referee #2 · 12 Oct 2018

This paper attempts to develop an estimate of the surface OA concentrations using three variables: (1) satellite HCHO column, (2) conversion factor from satellite HCHO column to surface HCHO concentrations (from a model), and (3) relationship between surface OA and HCHO concentrations derived from airborne measurements. The authors examined the relationship between surface OA and HCHO concentrations from a number of airborne measurements. They find that this relationship varies greatly among different studies (Table 1 and Figure 2). They further examine the dependence of OA-HCHO relationship on ambient NOx concentrations, and they do find that the slope gets lower with increasing NOx (Figure 3). The GEOS-Chem model was able to reproduce this relationship in the absence of wildfires. Finally, the authors use the relationship established from SEAC4RS to estimate surface OA and it compares well

with IMPROVE network. They found that adding NOx dependence or special treatment to urban cities do not change the correlation coefficient between OA estimate and IMPROVE OA. I have a few comments:

1. It seems that this relationship is largely driven by the relative contribution of POA vs. SOA to the total OA burden. As HCHO is mostly secondarily produced, this OA-HCHO is expected to have high slopes when OA emission is dominated by POA (such as wildfires shown in Figure 2). For SEAC4RS or DC3 this slope is much lower because OA is likely dominated by SOA. For KORUS-AQ or CalNex, this slope is likely driven by a mixture of biogenic and anthropogenic emissions, which falls somewhere between. It seems important to understand the contribution of POA vs. SOA to the total OA burden, before one uses this OA-HCHO to estimate OA. I am wondering if this can be investigated in their model.

2. Time dependence of OA-HCHO relationship. If indeed OA is dominated by SOA, one would expect a time-dependence of OA-HCHO relationship, as OA and HCHO are produced at different rates. How would this possibly affect their results?

3. Conversion from mid-day to daily average. The authors use the relationship between HCHO and OA derived from airborne measurements, and satellite HCHO column (overpass time is 130pm local time), to derive a surface OA, which should be the OA in local time 130pm. Then the authors compare that to IMPROVE monthly average data. It seems logical to add a correction factor from 1:30pm to daily or monthly average. The argument in Line 494-495 "ground OA in 494 the Southeast US were observed to have little diurnal variation" is not good enough.

4. The authors show in Figure 7. that different cases make little difference on correlation coefficients. Can the authors show how exactly the NOx dependence is implemented? What kind of NOx concentrations did they for IMPROVE sites (130pm or daily average)? Can the authors show how many large urban cities are treated differently in case 3 and 4? So the reader can see how many IMPROVE sites are affected by

this and understand why the effect is so small. Also the authors need to show how OA concentrations are affected as well.

5. Y-axis label of Figure 7 should be fixed.

---

## Author Comment (AC1) · 20 Nov 2018

**Responses to reviewers' comments on the paper of**
**"Towards a satellite – in situ hybrid estimate for organic aerosol abundance" by Jin Liao et a**l.

We thank the reviewers for their comments on our paper. To guide the review process we have copied the reviewer comments in black text. Our responses are in regular blue font. We have responded to all the referee comments and made changes **in bold text**.

**Anonymous Referee #1**

R1.0. This study aims to use the information in HCHO satellite observations, along with a model estimate of the vertical distribution, and in situ HCHO:OA relationships, to estimate surface OA concentrations over the United States. This is an interesting concept, though it is challenged with many uncertainties and assumptions and the manuscript is a bit unfocused. Here are the major issues that I have identified:

R1.1. The title is vague and somewhat inaccurate. The estimate involves satellite, in situ observations AND a model, the "hybrid" should be described as such. Given that the results presented are US only, "over the continental United States" should be specified in the title. And finally, I would recommend that the authors modify the title to include "formaldehyde" so that it is clear that this is not an estimate based off of satellite AOD.

We appreciate the reviewer's suggestion about the title.
To keep the title simple, "over the continental United States" is included in the abstract instead.

**In the abstract we added "over the continental US" in Line 46 as:**
**"Near-surface OA over the continental US are estimated by combining observed in situ relationships with HCHO column retrievals from NASA's Ozone Monitoring Instrument (OMI)."**

**In the abstract (L 48), we added OA estimate with η from satellite a priori profiles, as:**
**"HCHO vertical profiles used in OA estimates are from climatology a-priori profiles in the OMI HCHO retrieval or output for specific time period from a newer version of GEOS-Chem."**

**Line 48 – 53, we have changed "We evaluate this OA estimate against OA observations from the US EPA IMPROVE network and simulated OA from the GEOS-Chem global chemical transport model. The OA estimate compares well**

**with IMPROVE data obtained over summer months (e.g. slope = 0.62, r = 0.56 for August 2013), comparable to intensively validated GEOS-Chem performance (e.g. slope = 0.57, r = 0.56) and superior to the correlation with satellite-derived total aerosol extinction (r = 0.41)."**
**To:**

**"We evaluate these OA estimates against OA observations from the US EPA IMPROVE network and simulated OA from the GEOS-Chem global chemical transport model. The OA estimates compare well with IMPROVE data obtained over summer months (e.g. slope = 0.60-0.62, r = 0.56 for August 2013), comparable to intensively validated GEOS-Chem performance (e.g. slope = 0.57, r = 0.56) and superior to the correlation with satellite-derived total aerosol extinction (r = 0.41). This also indicates that OA estimates are not very sensitive to HCHO vertical profiles and that a priori profiles from OMI HCHO retrieval are similar to that from the newer model version in estimating OA."**

Because satellite HCHO data containing the modeling components we need for the organic aerosol estimate, we still keep using "satellite – in situ hybrid" in the title.

As suggested by the reviewer, we included formaldehyde in the title.

**We changed the title to be:**
**"Towards a satellite formaldehyde – in situ hybrid estimate for organic aerosol abundance".**

R1.2. What are the implications of the statement on lines 88-89 that "isoprene SOA is not the dominant source of SOA in summer" for the GEOS-Chem simulation in that region? The study refers several times to the GEOS-chem simulation for the SEUS being "extensively validated" (lines 52, 87, 239) but seems to refer to isoprene SOA. Does the simulation capture the non-isoprene components of SOA as well?

Thanks for pointing this out.
According to Figure 5 in Kim et al. (2015), this GEOS-Chem version (v9-02) reasonably well captured the non-isoprene components of SOA during SEAC[4]RS as well.
**To avoid confusion, we deleted line 88-89 "but this is only one location and it is likely that isoprene SOA is not the dominant source of SOA in summer."**

R1.3. The fire analysis (inset Figure 2) is not very convincing given the small number of points. Why did the authors not include data influenced by fire from other regions in their

analysis? In particular, the agricultural fire analysis appears to rely on very few points and the discussion of those results in lines 324-326 and 344-346 should be tempered.

Thanks for the reviewer's comment. Now we use specific term to constrain the statements.

**Lines 324-326: changed "OA exhibits a tight correlation with HCHO for both wildfires and agricultural fires". To "OA exhibits a tight correlation with HCHO for both wildfires and agricultural fires from SEAC$^4$RS data."**

**Line 326 added: "SEAC$^4$RS data are chosen because it sampled fires and had state-of-the-art, high quality measurements."**

**Lines 344-346: changed "The slope of OA to HCHO was higher for wildfires than agricultural fires. This may indicate that more OA is emitted in wildfires which often have higher intensity than agricultural fires (Liu et al., 2017; Forrister et al., 2015)."**
**To "The slope of OA to HCHO was higher for wildfires than agricultural fires during SEAC$^4$RS. This is consistent with more OA emitted in wildfires than agricultural fires (Liu et al., 2017). The factors driving higher OA to HCHO with wildfires are not clear and may be related to burning conditions and fuels."**

R1.4. The relationship between OA and HCHO seems to break down as you increase NOx (it's not all that strong to begin with, see point 7). The correlation coefficient of 0.44 implies under high NO2 less than 20% of the variability of OA can be explained by HCHO. An OA estimate based off of this weak relationship does not seem credible. Also: was NO2 not measured during CalNex and DC3 (Lines 176-183)? Why are these campaigns not parsed for NOx?

Thanks for pointing out the confusion. Ambient data are different from lab experiments that have controlled conditions or simulations that have known mechanisms. To be clearer to reviewer and readers, we modified the text as below.

**Line 376 added "The correlation coefficient of 0.45 for high NO$_2$ and isoprene conditions during SEAC$^4$RS is not very high but still shows significant dependence of the OA-HCHO relationship on the product of NO$_2$ and isoprene, considering that these are ambient data and other factors (e.g. different specific sources) also play a role in determining OA-HCHO relationships."**

**Line 383 changed "The OA / HCHO ratio clearly decreased as $NO_2$ levels increased during KORUS-AQ, suggesting that high NO conditions accelerated HCHO formation more than they did SOA production." To "The OA / HCHO ratio clearly decreased as $NO_2$ levels increased during KORUS-AQ, suggesting that high NO conditions accelerated HCHO formation more than they did SOA production. OA-HCHO relationships do not have dependence on local time of the day (not shown). This further confirms that $NO_x$ is an important factor that affects the OA/HCHO relationship."** This case is impressive for ambient data.

**Line 560-562: changed "As the in situ data showed a $NO_2$–isoprene-dependent OA and HCHO relationship, we attributed this to the uncertainty of isoprene emissions from MEGAN or IMPROVE network measurements." Into "As the in situ data showed a moderate $NO_2$–isoprene-dependent OA and HCHO relationship, we attributed this to the uncertainty in isoprene emissions from MEGAN, the locations of IMPROVE site at rural regions, the uncertainty in IMPROVE network measurements, or other factors (e.g. sources-dependent OA-HCHO) besides $NO_2$-isoprene that also need to be taken into account when determining the specific OA-HCHO relationship."**

**Line 388 added "Because OA and HCHO were tightly correlated during CalNex and DC3, we did not parse for $NO_x$. The $NO_x$ range during DC3 low altitude data was smaller than KORUS-AQ and SEAC⁴RS. DC3 OA and HCHO relationships only had slight dependence on $NO_2$ (not shown here), largely due to the limited dataset. The $NO_x$ range during CalNex low altitude data was large. The tight OA and HCHO correlation during CalNex could be due to the combination of different VOCs sources and $NO_x$ levels."**

R1.5. A key part of this analysis is the conversion of HCHO column to surface using the GEOS-Chem model. Are the AMFs used in the OMI product consistent with the eta's used here? How does the model treat boundary layer mixing? Are the results sensitive to this? Why do the authors only show SEAC4RS profiles of HCHO? How does the model perform for DC3 and CalNex? Lines 455-456 discusses the potential for BB plumes to impact the vertical profile; this could be investigated with the DC3 data.

Thanks for pointing out this. The satellite a priori profiles are used now. The text is modified as below.

To evaluate if the OA estimates are sensitive to η from different model versions, we also estimated OA using η from a priori profiles used in OMI retrieval AMF calculations.

**Line 267-269: changed "η(i) is derived from GEOS-Chem (v9-02), which includes updated isoprene scheme for OA and is the next version of the model (v9-01-03) for a priori profiles used in SAO satellite HCHO retrievals."**

**To " η (i) is derived from the HCHO a priori profiles used in SAO OMI air mass factor (AMF) calculations (GEOS-Chem v9-01-03 climatology) or from GEOS-Chem v9-02, which includes updated isoprene scheme for OA and is the next version of the model (v9-01-03) for a priori profiles used in SAO satellite HCHO retrievals. HCHO a priori profiles are used to be consistent with satellite HCHO retrievals and also to show that OA estimate can be derived without running a global model separately. The newer version of GEOS-Chem is used to test the sensitivity of OA estimates to updated version of η. The newer version of GEOS-Chem also allows sampling through the flight tracks of a recent field campaign (SEAC[4]RS) flight tracks and examining the factors impacting η with both modeled and measured HCHO profiles. The detailed information about the impact of HCHO profiles on η is provided in Sect. 5."**

**Line 490 added " η in Fig. 6a OA estimate is from GEOS-Chem v9-02 output for the specific month August, 2013. η in Fig. 6b OA estimate is from the climatology HCHO profiles satellite data used as a priori profiles from GEOS-Chem v9-01-03. The correlations of OA estimates with IMPROVE OA indicate that OA estimates are not very sensitive to η from different model versions"**

**Line 445 added "Because the sensitivity of OA estimates to η is investigated with η from different GEOS-Chem versions (Sect. 6.2), we don't compare HCHO vertical profiles from model and measurements for a comprehensive set of field campaigns. We chose SEAC[4]RS to ing the main factors impact the η over US because SEAC[4]RS has a larger spatial coverage than DC3 and CalNex."**

Line 455-456: changed "High concentrations of HCHO (e.g., in BB plumes) lofted by convection can also impact the vertical profile (Barth et al., 2015)." To "High concentrations of HCHO (e.g., in BB plumes) lofted by convection can also impact the vertical profile (Barth et al., 2015), **which is not further investigated because OA estimates with BB influences over US are excluded in current study**."

R1.6. The South Korea analysis is a null result and adds little to the paper. I suggest that the authors eliminate Section 6.7 and earlier analysis of KORUS-AQ data. They could re-cap in the Conclusions that a similar analysis was attempted for South Korea but was precluded by low HCHO from OMI.

Thanks for the reviewer's suggestion. We still want to keep the KORUS-AQ data analysis because the analysis provides useful information to indicate that this method enables estimating OA beyond continental US. To state the purpose of the South Korea analysis, we added the following text.

**Line 602, added "Although an OA estimate for South Korea could not be retrieved in the current study, the consistency in the dependence of OA-HCHO relationships on chemical factors (e.g. emission sources, NOx, and altitudes) provides important information for potential application of chemical factors dependent OA-HCHO relationships to the geographical domain beyond the continental US, especially with improved satellite HCHO data from Tropospheric Monitoring Instrument (TROPOMI)."**

R1.7. Need to develop a more detailed and quantitative discussion of uncertainties. Several potentially large uncertainties are discussed on lines 501-508 and Section 7. The authors should strengthen and consolidate this discussion to outline all the assumptions and potential impact on their analysis. In particular, I note that the correlations between HCHO and OA in the in situ observations are actually not all that high to begin with (in the case of SEAC4RS HCHO only explains 35% of the variability in OA).

Thanks for the comment. It is challenging to quantify the uncertainties of the OA estimates. For example, one of the uncertainty sources comes from satellite HCHO data. Zhu et al. (2016) had a detailed study about the comparison of different satellite HCHO retrievals with model results and in situ measurements. It is challenging to have an apple to apple comparison between satellite data with in situ measurements. The correlation coefficient between SAO HCHO retrieval and in situ measurements was only 0.24. To have a more detailed and quantitative discussion of uncertainties, we modified the text as below.

**Line 488 changed "The good correlation (correlation coefficient r = 0.56) between the OA estimate and corrected IMPROVE network measurements (Fig. 6e) indicates that the OA estimate can generally capture the variation of OA loading over the US." To**
**"Considering the uncertainties in satellite HCHO measurements, in using a campaign lump-sum OA-HCHO relationship to represent spatial resolved OA, in HCHO vertical profiles, and in ground IMPROVE network measurements, the correlation (correlation coefficient r = 0.56) between the OA estimate and corrected IMPROVE network measurements (Fig. 6f and 6g) is reasonably good**

**and indicates that the OA estimate can generally capture the variation of OA loading over the US."**

**Line 490 added "The correlation coefficient between HCHO SAO retrievals and in situ measurements during SEAC[4]RS was not high (r = 0.24) but this may be partly because they were not sampled at the same time. The uncertainty in HCHO SAO data was likely less than 76%. The uncertainty in applying a campaign lump-sum OA-HCHO relationship to individual spatial resolved satellite HCHO data to estimate OA induced an uncertainty of 41% according to the correlation coefficient of OA-HCHO in the field campaign. η in the Fig. 6a OA estimate is from a GEOS-Chem v9-02 output for the specific month August, 2013. η in the Fig. 6b OA estimate is the climatology from GEOS-Chem v9-01-03, the same as satellite data a priori profiles. The good correlations of OA estimates with IMPROVE OA indicate that OA estimates are not very sensitive to η from different model versions. The largest difference between the two OA estimates is their concentrations over East Texas. There are no IMPROVE OA measurements in the East Texas to evaluate which works better. The uncertainties in IMPROVE OA measurements, such as using a constant correction factor to correct the partial evaporation across all southeast US sites, and the spatially dependent OA/OC ratio (Tsigaridis et al., 2014), may also have contributed to the discrepancies between the OA estimate and EPA IMPROVE sites OA. Therefore, higher quality of satellite HCHO data and refining OA-HCHO relationships will help improve our OA estimate products. This combined with a spatially resolved IMPROVE OA correction factor and OA/OC ratios will help improve the correlation coefficients between OA estimates and IMPROVE OA."**

**Line 501-503 changed "Instead, the potential underestimation of HCHO from satellite retrieval (by 37%) (Zhu et al., 2016) compared to SEAC[4]RS may cause the low slope between the OA estimate and IMPROVE OA according to Eq. (1)." To "Instead, the potential underestimation of HCHO from satellite retrieval (by -37%) (Zhu et al., 2016) compared to SEAC[4]RS may be one of the most important reasons that cause the systematic difference (low slope) between the OA estimate and IMPROVE OA according to Eq. (1). Satellite HCHO data corrected by the low bias (by -37%) (Zhu et al., 2016) will increase our slopes of 0.60-0.62 to be close to the unity."**

**Line 501 added "The potential uncertainty (30%) in OA/OC ratio could also contribute to the systematic difference because we used OA/OC of 2.1 and studies (e.g. Pye et al., 2017; Canagaratna et al., 2015) showed that the OA/OC can range from 1.4 to 2.8."**

R1.8. Also many assumptions are based solely on the SEUS:

a. That OA accounts for a large fraction of submicron aerosol (lines 277-278). What about in other regions of the US?

Generally, OA is a major submicron aerosol component over the US.
**Line 277-278: added "and is one of the major submicron aerosol components over the US generally (Jimenez et al., 2009)."**

b. That the vertical profile of OA matches the total aerosol (line 284) – this has not been shown outside of SEUS and the potential for dust, nitrate, smoke plumes to alter this should be discussed

More information about the potential impact of dust, nitrate, and smoke plumes is added.

**Line 284: changed "The shape of average vertical profile of OA was very close to that of total aerosol mass (Wagner et al., 2015)" to "The shape of the average vertical profile of OA was very close to that of total aerosol mass for Southeast US (Wagner et al., 2015) where most of the enhanced aerosol concentrations over the US are located. Data with smoke plumes interferences are excluded in the following analysis. The potential contribution of dust and nitrate could alter the shape of the vertical profiles and introduce uncertainties when using OA vertical profiles for other parts of the US. Similar vertical profile shapes of OA and submicron particles were also observed in a campaign outside the US over South Korea (Nault et al., 2018). Though OA accounted for ~40% of the total submicron particles, the shape of OA and total submicron particles vertical profiles were nearly identical."**

**Line 536 added "The high surface aerosol extinctions (> 150 Mm$^{-1}$) (outliners in the scatter plot) are located in the Southeast US and therefore are not due to potential contribution of dust and nitrate altering the shape of vertical profiles outside of the SE US."**

c. Applying the OA:HCHO relationship from SEAC4RS for the entire US. The authors tested using data from the LA Basin as well, but how would OA (and HCHO) differ in the Midwest and NEUS? In Section 6.2 the authors discuss the ability of the OA estimate to

capture IMPROVE observations – how good is the estimate outside of the SEUS, where observations were used to construct the HCHO:OA relationship?

Thanks for the comment. The following discussions about Midwest and NEUS are added.

**Line 563 added "SEAC⁴RS and DC3 only had a few low altitude data in the Midwest and did not cover the Northeast US. The measured OA-HCHO relationship in the Midwest did not show significant difference from the SE US. The scatter plots (Fig. 6f and 6g) of OA estimates and IMPROVE OA do not show outliners for the Northeast and Midwest. This indicates that using the SEAC⁴RS lump sum OA-HCHO relationship can reasonably capture regions outside of the SEUS."**

MINOR

R1.9.    The abstract is missing a few key descriptors: that the analysis is performed for summertime only, that the estimate of near-surface OA is for the United States, and that that estimate also relies on the GEOS-Chem vertical profile of HCHO.

Thanks for the comment. The abstract is modified to include the information.
**Abstract line 46 changed "An estimate of near-surface OA is derived by combining observed in situ relationships with HCHO column retrievals from NASA's Ozone Monitoring Instrument (OMI)" to "An estimate of summer time near-surface OA over US is derived by combining observed in situ relationships with HCHO column retrievals from NASA's Ozone Monitoring Instrument (OMI). HCHO vertical profiles used in OA estimates are from climatology a-priori profiles in the OMI HCHO retrieval or output of specific period from newer version of GEOS-Chem.".**

R1.10. Lines 155-166: how do ISAF and DFGAS measurements compare during SEAC4RS?

As mentioned in the text, DFGAS measurements were available during DC3 but not SEAC⁴RS. **Line 162 added "HCHO measurements from ISAF also had a good agreement with DFGAS, with a correlation coefficient of 0.98 and a slope of 1.07.".**

R1.11. Line 174: specify what temperature was assumed for STP (273 and 298K are both used)

**Line 174 added "(at 273 K and 1013 mbar)".**

R1.12. Line 180: correct "for daytime NO"

Corrected

R1.13. Section 2.3: The spatial resolution of the products and and details of gridding and averaging are missing.

**Line 212 changed "Here we use the OMI HCHO version 2.0 (collection 3) retrieval" to "Here we use the OMI HCHO version 2.0 (collection 3) gridded (0.25 × 0.25) retrieval data".** The averaging information is provided in Line 218-219 "The monthly average HCHO columns were also weighted by the column uncertainties of the pixels."

R1.14. Lines 273-277: This sentence is awkwardly phrased ("can estimate the fraction of OA" and then the sentence says it can't do this...). Also why is MISR discussed here and later (lines 525-531) when it is not used in this study?

**Changed "The MISR satellite instrument can estimate the fraction of OA relative to total AOD, due to constraints on size range, shape and absorbing properties," to "The MISR satellite instrument can estimate a subset of AOD, due to constraints on size range, shape and absorbing properties"**

**Line 277 added "Because MISR estimates a subset of AOD, it is discussed here to verify we are not neglecting a satellite dataset that has already captured OA AOD."**

R1.15. Line 280: need to define Aext in the text

**Line 281: added "$A_{ext}$ is the calculated aerosol extinction (Mm$^{-1}$),"**

R1.16. Line 294: insert "near-surface in situ"

Done.

R1.17. Line 298: define "BB" in text

BB was defined in previous paragraph (Line 106)

R1.18. Line 402/Section 2.4: The emissions are not described in the model description section.

**Line 241 added "This model version uses the fourth-generation global fire emissions database (GFED4) (Giglio et al., 2013) as BB emission inventory."**

R1.19. Section 6.1: Are daily HCHO columns converted to surface concentrations using daily etas and then averaged to monthly values, or is the analysis performed using monthly means for everything?

This analysis is performed using monthly average satellite HCHO data.
**Line 463 changed "Satellite HCHO column" to "monthly average satellite HCHO column"**
**Line 463 changed "Satellite HCHO columns" to "Satellite monthly average HCHO column data."**
 **Line 464 added "either from climatology a priori profiles or monthly average HCHO profiles"**

R1.20. Line 488-9: A correlation coefficient of 0.56 is pretty modest. When only 31% of the variation is captured it is not accurate to state that the estimate "generally captured the variation" of the observations.

**Line 488-489:**
**Changed "The good correlation between the OA estimate and corrected IMPROVE network measurements (Fig. 6(e)) indicates that the OA estimate generally captured the variation of OA loading over the US." To "Considering the uncertainties in satellite HCHO measurements, in using the campaign lump-sum OA-HCHO relationship to represent spatial resolved OA, in HCHO vertical profiles, and in ground IMPROVE network measurements, the correlation (correlation coefficient r = 0.56) between the OA estimate and corrected IMPROVE network measurements (Fig. 6(e)) is reasonably good and indicates that the OA estimate can generally capture the variation of OA loading over the US."**

R1.21. Lines 537-538: The authors state their goal in using the extinction data here at the end of the Section. I suggest that the authors explain more clearly at the start of Section 6.3 why they are exploring extinction.

**Line 519-520: changed "To further evaluate the method of using satellite HCHO to derive an OA surface estimate, satellite measurements of AOD were converted to extinction for comparison." To "To further evaluate the method of using satellite**

**HCHO to derive an OA surface estimate, satellite aerosol measurements are used to approximate surface OA extinction for comparison. Satellite measurements of AOD were converted to surface extinction for the regions AOD is dominated by OA."**

---

## Author Comment (AC2) · 20 Nov 2018

**Responses to reviewers' comments on the paper of
"Towards a satellite – in situ hybrid estimate for organic aerosol
abundance" by Jin Liao et a**l.

We thank the reviewers for their comments on our paper. To guide the review process we have copied the reviewer comments in black text. Our responses are in regular blue font. We have responded to all the referee comments and made changes **in bold text**.

**Anonymous Referee #2**

R2.0. This paper attempts to develop an estimate of the surface OA concentrations using three variables: (1) satellite HCHO column, (2) conversion factor from satellite HCHO column to surface HCHO concentrations (from a model), and (3) relationship between surface OA and HCHO concentrations derived from airborne measurements. The authors examined the relationship between surface OA and HCHO concentrations from a number of airborne measurements. They find that this relationship varies greatly
among different studies (Table 1 and Figure 2). They further examine the dependence
of OA-HCHO relationship on ambient NOx concentrations, and they do find that the
slope gets lower with increasing NOx (Figure 3). The GEOS-Chem model was able
to reproduce this relationship in the absence of wildfires. Finally, the authors use the
relationship established from SEAC4RS to estimate surface OA and it compares well
with IMPROVE network. They found that adding NOx dependence or special treatment to urban cities do not change the correlation coefficient between OA estimate and IMPROVE OA. I have a few comments:

R2.1. It seems that this relationship is largely driven by the relative contribution of POA vs. SOA to the total OA burden. As HCHO is mostly secondarily produced, this OA-HCHO is expected to have high slopes when OA emission is dominated by POA (such as wildfires shown in Figure 2). For SEAC4RS or DC3 this slope is much lower because OA is likely dominated by SOA. For KORUS-AQ or CalNex, this slope is likely driven by a mixture of biogenic and anthropogenic emissions, which falls somewhere between. It seems important to understand the

contribution of POA vs. SOA to the total OA burden, before one uses this OA-HCHO to estimate OA. I am wondering if this can be investigated in their model.

We thank the reviewer for the comment. However, we believe that the situation is more complex, and that the OA-HCHO relationship is not just driven by the relative contribution of POA vs SOA to total OA.

For urban areas sampled from aircraft, recent analysis by Nault et al. (2018) indicates that the contribution of POA in KORUS-AQ is small (6-10%) and that the enhancement in OA is due to increased SOA production. This is consistent with past megacity studies (see references in that paper). Thus both OA and HCHO are dominantly secondary for the air masses sampled by the aircraft, and the slope is determined by their respective rates of production (see also Wood et al., 2010, which presents an analogous analysis for the OA vs Ox slope). At the relatively short photochemical ages sampled in KORUS-AQ, the difference in lifetimes for OA vs HCHO does not yet play a major role in changing the slope.

For biomass burning, POA is very large and the net formation of SOA is small (see e.g. Cubison et al., 2011). In this case the slope will be determined by the initial POA and HCHO emissions, together with SOA and especially HCHO production.

It is possible that in some cases the relative contributions of POA vs. SOA do play a controlling role on the OA/HCHO slope. This may become apparent as we analyze more cases.

**Line 334-335 changed "The high OA air masses also had high acetonitrile during KORUS-AQ." to "During KORUS-AQ, the high OA/HCHO air masses tended to have high acetonitrile. By the time we sampled, most organic aerosols were secondary (Nault et al., 2018). This indicates that the formation rates of OA and HCHO from different emission sources contribute to the different slopes of OA-HCHO. This also indicates that emission sources with enhanced acetonitrile tend to form more OA relative to HCHO downwind."**

The role of POA in impacting the slope of OA-HCHO in biomass burning case has been described in Line 339-343 "The slopes of OA vs. HCHO for BB air masses were higher than for anthropogenic and biogenic sources. This is consistent with high POA emission in BB conditions (Heald et al., 2008; Lamarque et al., 2010; Cubison et al., 2011), with low addition of mass due to

SOA formation (Cubison et al., 2011; Shrivastava et al., 2017)."

References:

Nault, B. A., Campuzano-Jost, P., Day, D. A., Schroder, J. C., Anderson, B., Beyersdorf, A. J., Blake, D. R., Brune, W. H., Choi, Y., Corr, C. A., de Gouw, J. A., Dibb, J., DiGangi, J. P., Diskin, G. S., Fried, A., Huey, L. G., Kim, M. J., Knote, C. J., Lamb, K. D., Lee, T., Park, T., Pusede, S. E., Scheuer, E., Thornhill, K. L., Woo, J.-H., and Jimenez, J. L.: Secondary Organic Aerosol Production from Local Emissions Dominates the Organic Aerosol Budget over Seoul, South Korea, during KORUS-AQ, Atmos. Chem. Phys. Discuss., https://doi.org/10.5194/acp-2018-838, in review, 2018.

Wood, E. C., Canagaratna, M. R., Herndon, S. C., Onasch, T. B., Kolb, C. E., Worsnop, D. R., Kroll, J. H., Knighton, W. B., Seila, R., Zavala, M., Molina, L. T., DeCarlo, P. F., Jimenez, J. L., Weinheimer, A. J., Knapp, D. J., Jobson, B. T., Stutz, J., Kuster, W. C., and Williams, E. J.: Investigation of the correlation between odd oxygen and secondary organic aerosol in Mexico City and Houston, Atmos. Chem. Phys., 10, 8947-8968, https://doi.org/10.5194/acp-10-8947-2010, 2010.

Cubison, M. J., Ortega, A. M., Hayes, P. L., Farmer, D. K., Day, D., Lechner, M. J., Brune, W. H., Apel, E., Diskin, G. S., Fisher, J. A., Fuelberg, H. E., Hecobian, A., Knapp, D. J., Mikoviny, T., Riemer, D., Sachse, G. W., Sessions, W., Weber, R. J., Weinheimer, A. J., Wisthaler, A., and Jimenez, J. L.: Effects of aging on organic aerosol from open biomass burning smoke in aircraft and laboratory studies, Atmos. Chem. Phys., 11, 12049-12064, https://doi.org/10.5194/acp-11-12049-2011, 2011.

R2.2. Time dependence of OA-HCHO relationship. If indeed OA is dominated by SOA,
one would expect a time-dependence of OA-HCHO relationship, as OA and HCHO are
produced at different rates. How would this possibly affect their results?

Thanks for the reviewer's comment. We understand that the OA-HCHO relationship could be time dependent, in principle, due to different lifetimes and different net chemical reaction rates. HCHO is close to being in steady-state with production rates roughly equal to loss rates while OA is not in steady-state with a

lifetime of a week.  Therefore, OA can be accumulated relative to HCHO when air masses are aged. This is evident as shown in Fig S2, using altitudes as an approximation of air mass age. Line 630-648 discussed the impact of the time dependent OA-HCHO relationship. The time dependent OA-HCHO relationship does not largely affect our study for monthly average OA over continental US because our OA estimates show reasonably good agreement with ground sites IMPROVE OA measurements. This also indicates that enhanced SOA are near the source regions statistically. We will consider the impact of air mass age (or time dependence of OA-HCHO relationship) when using refined OA-HCHO relationship to get higher spatial and temporal resolutions of OA estimate in the future.

**Line 634 added "HCHO is close to being in steady-state with production rates roughly equal to loss rates while OA is not in steady-state with a lifetime of a week.  Therefore, OA can be accumulated relative to HCHO when air masses are aged."**

**633-634 changed "OA vs. HCHO from SEAC[4]RS and KORUS-AQ field campaigns, color-coded with altitude, are plotted in Fig. S2 (a) and (b), respectively." To "OA vs. HCHO from SEAC[4]RS and KORUS-AQ field campaigns, color-coded with altitude as an indicator of air mass age, are plotted in Fig. S2 (a) and (b), respectively."**

**Line 648 added "Although OA-HCHO relationships depend on air mass age, it does not largely affect our study for monthly average surface OA over continental US because our OA estimates showed reasonably good agreement with ground sites IMPROVE OA measurements. This also indicates that SOA are enhanced near the source regions statistically. Nault et al. (2018) also showed the production of HCHO and SOA are similar and plateau around 0.5 - 1 photochemical day. So, in the near field of emissions and chemistry, the productions of these two species are similar; however, outside of near field of emissions and rapid chemistry, the long lifetime of OA vs the steady state of HCHO would start controlling the slopes and correlations."**

R2.3. Conversion from mid-day to daily average. The authors use the relationship be-
tween HCHO and OA derived from airborne measurements, and satellite HCHO column (overpass time is 130pm local time), to derive a surface OA, which should be the

OA in local time 130pm. Then the authors compare that to IMPROVE monthly average data. It seems logical to add a correction factor from 1:30pm to daily or monthly

average. The argument in Line 494-495 "ground OA in 494 the Southeast US were

observed to have little diurnal variation" is not good enough.

Thanks for pointing this out. We agree with the reviewer that it is important to estimate the impact of the inconsistency in using mid-day satellite HCHO data and daytime aircraft OA-HCHO relationships because HCHO has a clear diurnal profile and OA doesn't. Because the in situ flight data were collected in the daytime, the potential bias turns out to be small and we added as uncertainties in OA estimate. See the plot of the OA-HCHO relationships with and without HCHO corrected to 1: 30 pm according to the average HCHO diurnal profile. The text is modified as below.

[Figure]

**Line 497 added "However, surface HCHO has evident diurnal profiles with the highest concentrations around the mid-day (Kaiser et al., 2016), which could add uncertainties to OA estimate when using inconsistent time ranges of satellite HCHO data measured in the mid-day and in situ airborne OA and HCHO relationships measured in the daytime. The SEAC[4]RS HCHO concentrations were converted to 1:30 pm concentrations according to the average HCHO diurnal profile from the Southern Oxidant and Aerosol Study (SOAS) (Kaiser et al., 2016). The OA-HCHO relationship with HCHO converted to 1:30 pm yielded a slope of 5% lower than the original OA-HCHO relationship."**

**Line 497-501 deleted. "The difference in mid-day and daytime HCHO concentrations is not prominent, depends on the location and may contribute to a small bias to the mid-day OA estimate (DiGangi et al., 2012). This is probably due to increased boundary layer height diluting the photochemical formation of HCHO in the mid-day."**

R2.4. The authors show in Figure 7. that different cases make little difference on corre-
lation coefficients. Can the authors show how exactly the NOx dependence is imple-
mented? What kind of NOx concentrations did they for IMPROVE sites (130pm or daily
average)? Can the authors show how many large urban cities are treated differently
in case 3 and 4? So the reader can see how many IMPROVE sites are affected by
this and understand why the effect is so small. Also the authors need to show how OA
concentrations are affected as well.

Thanks for the comment. To be clearer, more detailed information is added.

**Line 556 added " The details about how to implement chemical factors dependent OA estimates for the four cases are also provided in Table 2. Satellite OMI $NO_2$ data (at 1 : 30 pm) are used to represent $NO_x$ levels and big cities are defined as $NO_2 > 4 \times 10^{15}$ molec $cm^{-2}$, the CalNex in situ OA-HCHO relationship is applied for big cities. It turns out that only 1 IMPROVE site (SAGA1) near LA was affected by high $NO_2$ and led to the insignificant change in case 3 compared to case 1. This is not unexpected because IMPROVE sites are in rural regions. The OA estimate in SAGA1 decreased from 1.88 ug m-3 from case 1 to 0.17 ug m-3 in case 3 while the measured OA in IMPROVE SAGA1 was 1.52 ug m-3. This may infer that CalNex is not very consistent with SEAC[4]RS due to different sampling instruments, strategies and seasons. Lowing the $NO_2$ threshold when defining big cities did not help improve the agreement either."**

Because case 3 only changed 1 site compared to case 1, case 4 was similar to case 2 as expected and did not improve the agreement between OA estimate and IMPROVE OA.

R2.5. Y-axis label of Figure 7 should be fixed.

Thanks for pointing this out. **It is changed to "Correlation coefficients of OA estimate v.s IMPROVE OA".**

---

## Author Response (AR2)

**Responses to reviewer #1's comments on the revised paper of "Towards a satellite – in situ hybrid estimate for organic aerosol abundance" by Jin Liao et a**l.

We thank the reviewer for his/her comments on our paper. We have copied the reviewer comments in black text. Our responses are in regular blue font. We have responded to all the referee comments and made changes **in bold text**.

The manuscript is improved over the original submission, though some aspects of the analysis remain poorly described. I've provided suggestions below to address these. Given all the uncertainties and the simplistic application of highly-variable source-specific relationships (lines 364-366 encapsulate the fundamental problem that makes it hard to justify any sort of universal application of slopes), I don't think that the authors have made a convincing case that this approach is generally worth pursuing, but given the work that went into the analysis, I think that the paper meets the bar for publication (once they have addressed the remaining issues).

We agree that the simplistic application of highly variable source specific relationships induce uncertainties in estimating OA. However, we can extract the common factors that govern the different OA-HCHO relationships and refine the OA-HCHO relationships to reduce the uncertainties. For example, the source dependent OA-HCHO relationships (Fig. 2) showed higher OA-HCHO slope in biomass burning and anthropogenic sources with inefficient combustions (e.g. KORUS-AQ) compared to biogenic and clean anthropogenic sources. The in situ data indicate that inefficient combustions contribute to the high slopes of OA-HCHO, probably due to both enhanced primary OA and increased formation of SOA. Therefore, it is possible to have a universal application of sources and chemical factors dependent slopes but more analysis is needed and it is beyond the scope of this paper.

To be clear, line 381 added

**"Overall, the source dependent OA-HCHO relationships (Fig. 2) showed highest OA-HCHO slopes of BB and heavily polluted anthropogenic sources with inefficient combustion (e.g., KORUS-AQ) compared to biogenic and relatively clean anthropogenic sources. This indicated that inefficient combustions contribute to the high slopes of**

**OA-HCHO, probably due to both enhanced primary OA and increased formation of SOA. Enhanced pre-existing aerosols such as primary aerosols can provide more surfaces to increase VOCs condensation and SOA formation. VOCs co-emitted from heavily polluted anthropogenic sources can also form more SOA. It is possible to extract the factors that govern the different OA-HCHO relationships and potentially have a universal application of the slopes as a function of the factors (e.g., sources and combustion efficiencies)."**

**Line 358-381 is also re-organized to show that we can extract the factors (e.g. sources) that govern the OA-HCHO relationships.**

1. Lines 105-107: It is unclear what the reader is meant to take from this sentence. Does this approach from de Vries et al. work well for OA? Are there major flaws? Are you trying to provide an alternative approach?
The approach from de Vries et al. characterized the aerosol types (e.g. OA) for AOD. Our approach is trying to quantify OA mass concentrations.

**Line 107 added**
**"Here we aim to provide a quantitative estimation of OA mass concentrations from satellite measurements."**

2. Lines 127-128: Jimenez et al. (2009) do not show that OA is a major contributor to AOD (they do not discuss AOD) – an alternate reference is required for this statement.

**line 127-128 changed "This also suggests that OA, a major contributor to AOD in the above cases (Jimenez et al., 2009), and HCHO share common emission sources and photochemical processes." To "This also suggests that OA share common emission sources and photochemical processes with HCHO and are an important contributor to AOD in the above cases."**

3. Line 180: state in the text why you convert from STP units

**line 180 changed "The OA measurements are from 1 min merge data and converted from µg sm$^{-3}$ (at 273 K and 1013 mbar) to µg m$^{-3}$ under local T & P for each data point." To "The OA measurements are from 1 min merge data and converted from µg sm$^{-3}$ (at 273 K and 1013 mbar) to µg m$^{-3}$ under local T & P for each data point, to be consistent with**

**HCHO concentrations in µg m$^{-3}$ or molec cm$^{-3}$ at local T & P.”**

4. Lines 182-189: give measurement details for NO2 for DC3 and CalNex here (you discuss these measurements on lines 429-432). Also need to describe source of isoprene measurements for SEAC4RS.

**line 189 added “SEAC$^4$RS isoprene measurements were from proton-transfer-reaction mass spectrometer (PTR-MS) (Wisthaler et al., 2002).**

**Reference:**
**Wisthaler A., Hansel A., Dickerson R. R., Crutzen P. J.: Organic trace gas measurements by PTR-MS during INDOEX 1999,  J Geosphys Res-Atmos, 107(D19), 8024, 2002.**

5. Lines 230-234: also state that using monthly product
**line 230 changed “Here, we use collection 06 (MYD04_L2, hbps://ladsweb.nascom.nasa.gov), retrieved using the Dark Target (DT) and Deep Blue  (DB) algorithms (Levy et al., 2015).” to “Here, we used collection 06 (NASA MODIS AOD data archive), retrieved using the Dark Target (DT) and Deep Blue  (DB) algorithms (Levy et al., 2015), monthly average data.**

**6. Lines 260-263: give global emissions of isoprene and NOx used in the model**

**line 260 changed “Global isoprene emissions are used to calculate an isoprene and NO$_2$ dependent OA estimate. Global isoprene emissions are from the Model of Emissions of Gases and Aerosols from Nature version 2.1 (Guenther et al., 2006) as implemented in GEOS-Chem and driven with MERRA (MEGAN-MERRA).” To “Global isoprene emissions from the Model of Emissions of Gases and Aerosols from Nature version 2.1 (Guenther et al., 2006) (MEGAN) and satellite NO$_2$ column data were used to calculate an isoprene and NO$_x$ dependent OA estimate (see Table 2). Global isoprene emissions from MEGAN were implemented in GEOS-Chem and driven with MERRA (MEGAN-MERRA).”**

**Also see line 654 added “OMI NO$_2$ column observations were used to represent surface NO$_2$ levels and surface isoprene emissions from**

**MEGAN were used to represent surface isoprene concentrations, assuming that $NO_2$ column observations reflect surface $NO_2$ distributions and isoprene emissions reflect the concentrations of isoprene due to its short lifetime (~1 hr). The detailed implementation is provided in the notes in Table 2. "**

**line 227-229: changed "Satellite $NO_2$ column observations are also derived from NASA's OMI level 3 data, archived at hbps://disc.sci.gsfc.nasa.gov as "OMI-Aura_L3-OMNO2d" (Lamsal et al., 2014)." To "Satellite $NO_2$ column observations were also derived from NASA's OMI level 3 data (Lamsal et al., 2014; NASA OMI $NO_2$ data archive). Satellite $NO_2$ observations were used to calculate $NO_x$ related chemical factor dependent OA estimate (see Table 2)."**

7. Line 271: mid-level of MERRA-2 surface layer is ~60m, not 140m Mid-level height of the surface layer should be used here.

Thanks for pointing out this.
**line271 changed "$\eta(i)$ is the ratio of midday surface layer (~140 m)" to "$\eta(i)$ is the ratio of midday surface layer (~60 m)"**

8. Line 302: concentrations at STP or under ambient conditions?
**line 302 changed "surface layer OA concentrations ($\mu g\ m^{-3}$) to column OA concentrations ($\mu g\ m^{-2}$)" to "surface layer OA concentrations ($\mu g\ m^{-3}$, at ambient T & P) to column OA concentrations ($\mu g\ m^{-2}$)".**

9. Lines 305: this is a false statement. Highest PM2.5 levels over the United States are generally in California and in urban regions (see "Our Nation's Air" report here: https://www.epa.gov/air-trends). Please correct.

**line 305 changed "The shape of the average vertical profile of OA was very close to that of total aerosol mass over SE US (Wagner et al., 2015) where most of the enhanced aerosol concentrations over the US are located" to "The shape of the average vertical profile of OA (OA fraction: 0.54-0.7) was close to that of total aerosol mass over SE US (Wagner et al., 2015) where a large fraction of the enhanced non-BB aerosol concentrations in summer time over the US are located."**

10. Figure 1: caption indicates that this shows the data used in this study, but the text indicates that only data < 1km was used. Please modify the figure to show only data below 1 km altitude.

Changed figure 1 as suggested

[Figure]

11. Figure 3a: a character is displaying incorrectly on the color bar label.
Thanks for pointing this out. The character is corrected.

[Figure]

12. Section 3.1: The authors did not address my concern about the low number of points used to characterize the HCHO:OA from fires. To explicitly address this please add the number of points used for each category/campaign to Table 1 so that this is clear to the reader. Please also add the caveat "though data is limited" at the end of the sentence "…fires during SEAC4RS." on line 376.

**line 375-376 changed "The slope of OA to HCHO was higher for wildfires than agricultural fires during SEAC[4]RS." To "The slope of OA to HCHO was higher for wildfires than agricultural fires during SEAC[4]RS though data were limited (see Table 1)."**

| | US (SEAC[4]RS) | US (DC3) | US (CalNex) | South Korea (KORUS-AQ) | Wild Fires (SEAC[4]RS) | Agricultural Fires (SEAC[4]RS) | SEAC[4]RS Low NO2 and Isoprene | SEAC[4]RS high NO2 and Isoprene |
|---|---|---|---|---|---|---|---|---|
| **In situ measurements OA v.s. HCHO** | | | | | | | | |
| Slope [a] | 1.93 ±0.07 | 1.30± 0.10 | 1.34 ±0.02 | 2.75±0.05 | 25.08±0.30 | 3.22±0.37 | 2.39±0.09 | 1.45 ±0.19 |
| Slope [b] (×10[-11]) | 9.61 ± 0.34 | 6.49± 0.49 | 6.66 ± 0.09 | 13.7  1 ± 0.25 | 125  .05 ± 1.49 | 16.04 ± 1.85 | 11.9±043 | 7.25 ± 0.96 |
| Intercept[c] | 0.34 ±032 | 1.10±0.30 | -0.90 ±0.06 | 1.36±0.22 | −6.85±2.80 | 10.41±5.82 | −1.14 ±0.37, | 1.14 ±1.22 |
| Correlation coefficient r | 0.59 | 0.76 | 0.88 | 0.70 | 0.97 | 0.85 | 0.64 | 0.45 |
| Number of points (1 min avg) | 1506 | 134 | 1772 | 3425 | 515 | 32 | 1138 | 226 |
| **GEOS- Chem model sampled along the flight track OA v.s. HCHO** | | | | | | | | |
| Slope[a] | 1.25 ±0.03 | | | 1.39±0.05 | 0.48±0.05 | | | |
| Slope (×10[-11]) | 6.21 ± 0.14 | | | 6.95± 0.23 | 2.37± 0.22 | | | |
| Intercept | −1.32 ±0.11 | | | 1.88 ± 0.07 | 0.12±0.03 | | | |
| Correlation Coefficient r | 0.76 | | | 0.43 | 0.53 | | | |

13. Line 432: you state that the NOx range during CalNex was large, but you do not indicate whether the HCHO:OA showed a relationship with NOx in this region. Please address.

**Line 432 changed "The tight OA and HCHO correlation during CalNex could be due to the combination of different VOCs sources and $NO_x$ levels." To "The OA and HCHO correlation during CalNex was very tight and the slope of OA-HCHO did not show clear dependence on $NO_x$, which could be due to the combination of different VOCs sources and $NO_x$ levels."**

14. Section 6.1: why was this analysis performed on the monthly time scale? Would a daily analysis have any benefits? Please discuss.

**Line 524 added "The OA estimate was calculated on the monthly time scale, largely because OA estimate is based on OMI HCHO observations and uncertainty weighted average for a time scale of about one month (Gonzalo et al., 2015; Zhu et al., 2016) is needed to reduce the noise in daily OMI HCHO data. With improved satellite HCHO data from TROPOMI, higher time resolution (e.g. weekly average) HCHO data could be useful to estimate OA in the future."**

15. Line 522: you used (NOx)(isoprene) to define the chemical conditions so this is what you should specify here, not "NOx levels" alone.

**Line 522: changed (e.g., NOx levels) to (e.g., $NO_x$ and isoprene levels)**

16. Line 527: specify in text what slope is used for Case 1
**Line 527 changed "The monthly average surface OA estimate over the US in August 2013 for case 1" to "The monthly average surface OA estimates over the US in August 2013 using SEAC[4]RS lump-sum slope and intercept"**

17. Line 569: remove "slightly"
**Removed.**

18. Line 604: Section 2.6 does not describe what fraction of AOD comes from OA; how was this determination made here?
**line 603-604: deleted "for the regions AOD is dominated by OA"**

19. Line 629-630: It should be noted in the text that the "GEOS-Chem simulation" and the "OA estimate" are not independent.

**line 629-630 changed "The GEOS-Chem simulation had a coarser resolution than satellite HCHO data." To "Although HCHO vertical profiles from GEOS-Chem were used in OA estimate, the GEOS-Chem simulation had a coarser resolution than OA estimate."**

20. Line 640: the authors should briefly explain the four cases in the text to start Section 6.5.

**line 640 added: "OA were estimated with different OA-HCHO relationships for 4 cases (Table 2). LUMP-SUM was using the non-BB SEAC$^4$RS campaign lump-sum relationship, the same as shown in Fig. 6; ISOP-NOx was using non-BB SEAC$^4$RS NO$_2$ and isoprene dependent relationship; URBAN was using CalNex for large urban cities and SEAC$^4$RS lump-sum for other US regions; and COMBINE was using CalNex for large urban cities and NO$_2$ and isoprene dependent non-BB SEAC$^4$RS for other US regions."**

21. Lines 643-645: the authors indicate that they use NO2 observations from OMI data. What is the source of the isoprene concentration data needed to apply the (NOx)(isoprene) chemical characterization in case 2 and 4? Is this from the GEOS-Chem model? If so, some comparison of how well GEOS-Chem reproduces the observed (NOx)(isoprene) during SEAC4RS should be added to the manuscript.

See the notes in Table 2 and line 260-263 about the representation of isoprene in chemical characterization in case 2 (ISOP-NOx) and case 4 (COMBINE). We aim to use spatiotemporal resolved global dataset (e.g. satellite data or emission inventories) to represent the chemical factor. Isoprene emissions instead of isoprene concentrations are used in chemical characterization. Because the lifetime of isoprene is sufficiently short (~ 1hr), isoprene emissions can generally reflect the isoprene concentrations. Isoprene emission inventory is also easier to access and used by global models to simulate isoprene concentrations. Similarly based on the short lifetime of isoprene, previous studies (e.g., Palmer et al., 2003) have used satellite column concentrations to derive isoprene emissions. Global

isoprene emissions are from MEGAN. See line 261-263: Global isoprene emissions are from the Model of Emissions of Gases and Aerosols from Nature version 2.1 (Guenther et al., 2006) as implemented in GEOS-Chem and driven with MERRA (MEGAN-MERRA)."

Reference:
Palmer P. I., Jacob D. J., Fiore A. M., and Martin R. V.: Mapping isoprene emissions over North America using formaldehyde column observations from space, J Geophys Res-Atmos, 108(D6), 4180, 2003.

**To be clear, line 654 added "OMI $NO_2$ column observations were used to represent surface $NO_2$ levels and surface isoprene emissions from MEGAN were used to represent surface isoprene concentrations, assuming that $NO_2$ column observations reflect surface $NO_2$ distributions and isoprene emissions reflect the concentrations of isoprene due to its short lifetime (~1 hr). The detailed implementation is provided in the notes in Table 2. "**

22. Line 780-781: the authors should also indicate that the OA estimate is biased low.

**line 780-781: changed "The OA estimate over the continental US generally correlated well with EPA IMPROVE network OA measurements corrected for partial evaporation." to "The OA estimate over the continental US generally correlated well with EPA IMPROVE network OA measurements corrected for partial evaporation, with a biased low slope of 0.62 or 0.60, mostly due to underestimation of HCHO concentrations from the OMI HCHO retrieval."**

23. The manuscript includes many small grammatical errors and some awkward phrasing; it should be edited for language

Edited.